# FLOW POLICY GRADIENTS FOR LEGGED ROBOTS

## ABSTRACT

We study robot control with flow policy optimization (FPO), an online reinforcement learning algorithm for flow-based action distributions. We demonstrate how flow matching policies can be trained from rewards for more difficult continuous control tasks than shown in prior work, using a set of design choices that reduce gradient variance and regularize entropy. We show that these design choices mitigate policy collapse challenges faced by the original FPO algorithm and use the resulting algorithm, FPO++, to train flow policies for legged robot locomotion and humanoid motion tracking. We find that FPO++ is stable to train, interpretably models cross-action correlations, and can be deployed to real humanoid robots. Sim2real video results can be found on our anonymous webpage.[1]

## 1 INTRODUCTION

Recent work in Flow Policy Optimization (FPO) (McAllister et al., 2025) has demonstrated how flow matching models (Lipman et al., 2023) can be trained in an online, policy gradient-based reinforcement learning setting. Flow-based policies are attractive because they generalize both simple and complex continuous action distributions, while remaining simple to implement. They are therefore promising for continuous control in robotics, both for fine-tuning flow-based policies learned via behavior cloning (Black et al.; Chi et al., 2024a) and for training flow policies from scratch.

Despite promising performance in synthetic benchmarks, we found it challenging to naively apply flow policies to real robot control challenges. In this work, we therefore introduce FPO++: an improved version of FPO that is stable and effective for real-world robot control problems. We document challenges associated with training standard FPO policies on robot locomotion and motion tracking tasks—notably, both sudden and gradual policy collapse—and show that a small but critical set of algorithmic changes mitigates these problems. Specifically, FPO++ proposes (1) an updated likelihood ratio approximation that increases effective batch size, (2) an entropy-preserving trust region objective inspired by DAPO (Yu et al., 2025) and SPO (Xie et al., 2024), and (3) numerically stable CFM loss computation.

We evaluate FPO++ on a diverse set of robotic tasks across four simulated robots (Unitree Go2, Boston Dynamics Spot, Unitree H1, and Unitree G1), demonstrating stable training on quadrupedal and bipedal locomotion benchmarks as well as humanoid motion tracking. Our experiments show that FPO++ is significantly stabler to train than standard FPO, and can achieve competitive performance when compared to Gaussian PPO baselines. We analyze the learned policy distributions, which reveal that FPO++ captures interpretable cross-action correlations during training. We further validate these results through zero-shot sim-to-real transfer, deploying FPO++ policies trained entirely in simulation to two physical humanoid robots (Unitree G1 and Booster T1). To facilitate further research, we will provide open-source implementations of FPO++ along with training configurations for all tasks.

## 2 IMPROVED FLOW POLICY OPTIMIZATION

We introduce FPO++: an updated version of the FPO algorithm that stabilizes training and improves performance in challenging, real-world robotics tasks. To present FPO++, we first summarize the flow matching policy gradient framework. We then discuss the failure modes we observed in naive FPO implementations, followed by specific updates made in FPO++.

---

[1] Project webpage: `https://fpocontrol.github.io/`

## 2.1 PRELIMINARIES

**Policy Gradients and PPO.**   In on-policy reinforcement learning, rollouts in the form of per-timestep observation, action, and reward tuples $(o_t, a_t, r_t)$ from a policy $\pi_\theta(a_t \mid o_t)$ are used to update the policy to maximize expected return. The dominant approach for achieving this is *Proximal Policy Optimization* (PPO) (Schulman et al., 2017), which applies an on-policy trust region using a clipped likelihood ratio:

$$\max_\theta \ \mathbb{E}_{a_t \sim \pi_{\theta_{\text{old}}}(a_t \mid o_t)} \left[ \min\left( r(\theta)\hat{A}_t, \ \text{clip}(r(\theta), 1 - \varepsilon^{\text{clip}}, 1 + \varepsilon^{\text{clip}})\hat{A}_t \right) \right], \tag{1}$$

where $\hat{A}_t$ is an advantage estimate (Schulman et al., 2015b) and $r(\theta)$ is the likelihood ratio,

$$r(\theta) = \frac{\pi_\theta(a_t \mid o_t)}{\pi_{\theta_{\text{old}}}(a_t \mid o_t)}. \tag{2}$$

PPO is popular because it is simple to implement and provides strong empirical performance. It also inherits the advantages of general policy gradient methods, requiring differentiability only from action likelihoods and not from a reward model or environment dynamics.

**Flow Policy Optimization (FPO).**   The goal of the FPO (McAllister et al., 2025) algorithm is to enable PPO-style training of policies parameterized as flow models (Lipman et al., 2023). While likelihoods under the distribution captured by a flow model can be estimated (Skreta et al., 2025), doing so in a reinforcement learning setting is computationally prohibitive. To address this, FPO replaces the PPO likelihood ratio $r(\theta)$ with a surrogate,

$$\hat{r}_{\text{FPO}}(\theta) = \exp\left( \hat{\mathcal{L}}_{\text{CFM},\theta_{\text{old}}}(a_t; o_t) - \hat{\mathcal{L}}_{\text{CFM},\theta}(a_t; o_t) \right), \tag{3}$$

where $\hat{\mathcal{L}}_{\text{CFM},\theta}(a_t; o_t)$ is a Monte Carlo estimate of the conditional flow matching (CFM) loss.

This formulation enables PPO-style training of expressive flow-based policies, and can be applied in a way that mirrors PPO's clipped objective:

$$\max_\theta \ \mathbb{E}_{a_t \sim \pi_\theta(a_t \mid o_t)} \left[ \min\left( \hat{r}_{\text{FPO}}(\theta)\hat{A}_t, \ \text{clip}(\hat{r}_{\text{FPO}}(\theta), 1 - \epsilon, 1 + \epsilon)\hat{A}_t \right) \right]. \tag{4}$$

Intuitively, FPO's ratio approximation uses CFM loss differences to approximate action log-likelihood differences; as discussed by McAllister et al. (2025), this construction can be justified by interpreting the CFM loss as a variational bound. The final objective (Equation 4) then uses advantage estimates to shift probability flow toward higher-reward actions.

**Conditional flow matching loss.**   To estimate CFM losses, FPO first draws $N_{\text{mc}}$ random noise $\epsilon_i \sim \mathcal{N}(0, I)$ and flow step $\tau_i \in [0, 1]$ samples for each action $a_t$. Multiple noised actions are then computed using an interpolation schedule defined by $\tau_i$,

$$a_t^{\tau_i} = (1 - \tau_i)a_t + \tau_i \epsilon_i \tag{5}$$

Squared errors are computed and averaged for the policy's velocity predictions $\hat{v}_\theta$,

$$\hat{\mathcal{L}}_{\text{CFM},\theta}(a_t; o_t) = \frac{1}{N_{\text{mc}}} \sum_i^{N_{\text{mc}}} \ell_\theta(a_t, \tau_i, \epsilon_i; o_t) \tag{6}$$

$$\ell_\theta(a_t, \tau_i, \epsilon_i; o_t) = \| \hat{v}_\theta(a_t^{\tau_i}, \tau_i; o_t) - (a_t - \epsilon_i) \|_2^2. \tag{7}$$

These losses can then be used in the FPO ratio approximation (Equation 3) for flow policy updates, which aims to decrease CFM losses for actions with positive advantages and increase CFM losses for actions with negative advantages.

While the standard FPO formulation succeeds in synthetic benchmarks (McAllister et al., 2025), we found that it required refinements to achieve reliable performance in more difficult tasks.

## 2.2 FLOW POLICY FAILURE MODES

We observed that naive FPO implementations have two common failure modes when applied to more difficult robot control tasks. We show examples of these failure modes in Figure 1, and provide descriptions below.

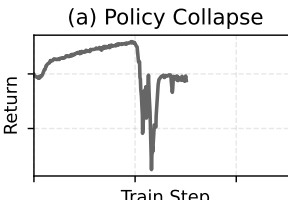 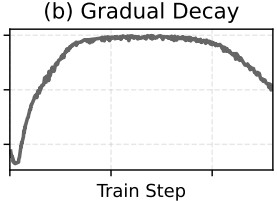 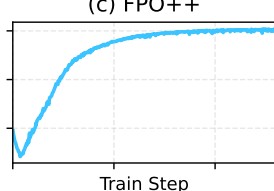

Figure 1: **FPO failure modes.** We found that naively applying FPO to more difficult reinforcement learning tasks often results in instabilities during training. *(a)* Sudden collapse in episode returns. *(b)* Fast initial learning, followed by gradual decay. *(c)* Stable training using FPO++.

**Policy collapse during training.** FPO implementations that achieve high rewards on DMC tasks (Tassa et al., 2018) encounter frequent instabilities when applied to more challenging robot locomotion tasks. This can be characterized by large drops in average training returns. Instabilities would irrecoverably halt training across tasks, even after tuning hyperparameters like learning rate, clip threshold, weight decay, and normalization strategies. Examples can be observed across hyperparameter choices in Figure 2.

**Gradual decay after returns peak.** As policy learning progresses, we hope to see steady increases in average training returns. This should happen until an optimal policy is found; afterwards, policy performance should plateau. While initial FPO rewards often peaked faster than PPO rewards, we found that policy performance sometimes began to decay after this peak. We observed that this happens after entropy collapse in FPO policies. If the policy's entropy is too low to explore effectively, rewards begin to decay because each sampled action carries a small approximation error from Euler integration, which accumulates across policy updates.

## 2.3 FPO++

FPO++ proposes a set of changes to FPO for (1) reducing the variance of gradients during training, (2) regularizing entropy of action distributions, and (3) improving numerical stability. We find that these changes mitigate the failure modes discussed in Section 2.2, while improving overall performance for converged policies.

**Increasing effective batch size.** Unlike Gaussian policies that compute a single likelihood per action, FPO estimates CFM losses by averaging over multiple $(\tau_i, \epsilon_i)$ samples. In the standard FPO framework (McAllister et al., 2025), this average is performed before the exponential, producing a single ratio per action:

$$\hat{r}_{\text{FPO}}(\theta) = \exp\left( \frac{1}{N} \sum_{i=1}^{N} \left( \ell_{\theta_{\text{old}}}(a_t, \tau_i, \epsilon_i; o_t) - \ell_\theta(a_t, \tau_i, \epsilon_i; o_t) \right) \right). \quad (8)$$

In the context of PPO-style clipping, an important characteristic of this formulation is that ratios are clipped *after* averaging across samples. For a given action, this means that either all or no samples are clipped. In FPO++, we instead compute ratios on a per-sample basis:

$$\hat{r}_{\text{FPO}}^{(i)}(\theta) = \exp\left( \ell_{\theta_{\text{old}}}(a_t, \tau_i, \epsilon_i; o_t) - \ell_\theta(a_t, \tau_i, \epsilon_i; o_t) \right). \quad (9)$$

Each $(\tau_i, \epsilon_i)$ pair contributes its own ratio, with the same advantage $\hat{A}_t$ shared across all samples.

These two ratio formulations produce identical gradients for on-policy data, where all ratios evaluate to 1. When taking multiple gradient steps on the same batch of transitions, however, the per-sample ratio provides a finer-grained trust region than the original per-action formulation. This leads to higher and more stable final policy returns.

**Entropy-preserving trust region (ASPO).** We found that the stability of FPO training can be improved significantly by adjusting its trust region implementation. In FPO++, we adopt an asymmetric trust region inspired by Yu et al. (2025) that we refer to as *Asymmetric SPO (ASPO)*. We use

standard PPO clipping positive advantages; for negative advantages, we adopt the more constrained Simple Policy Optimization (SPO) objective proposed by Xie et al. (2024):

$$\max_{\theta} \quad \mathbb{E}_{a_t \sim \pi_{\theta_{\text{old}}}(a_t|o_t)} \left[ r(\theta)\,\hat{A}_t \;-\; \frac{|\hat{A}_t|}{2\,\varepsilon^{\text{clip}}} \left( r(\theta) - 1 \right)^2 \right] \tag{10}$$

Like the asymmetric design proposed by (Yu et al., 2025), the SPO objective we use for negative advantages preserves entropy in the action distribution by providing gradient signals that discourage aggressive likelihood decreases. We find empirically that this is critical for stability in FPO++.

The SPO objective also reduces gradient variance. PPO clipping zeros out gradients for samples that pass the trust region, which leads to increasingly sparse and noisy updates. In contrast, the SPO objective retains gradients for all samples that it is applied to (Xie et al., 2024).

**Improving numerical stability.** The FPO surrogate ratio (Eq. 3) involves exponentiating differences of squared CFM losses. We found this operation to be the source of numerical problems in FPO: loss outliers in the $(\tau_i, \epsilon_i)$ sampling process can easily cause instabilities after being squared and then exponentiated. We address this with two steps. First, we replace the L2 conditional flow matching objective with a robust Huber loss:

$$\ell_\theta^{\text{Huber}}(a_t, \tau_i, \epsilon_i; o_t) = \rho_\delta \left( \hat{v}_\theta(a_t^{\tau_i}, \tau_i; o_t) - (a_t - \epsilon_i) \right), \tag{11}$$

where the Huber kernel $\rho_\delta$ is defined as

$$\rho_\delta(x) = \begin{cases} \frac{1}{2}\|x\|_2^2, & \text{if } \|x\|_2 \leq \delta, \\ \delta\left(\|x\|_2 - \frac{\delta}{2}\right), & \text{if } \|x\|_2 > \delta. \end{cases} \tag{12}$$

Second, we apply a gradient-preserving clamping operator $\phi$ to the CFM loss difference:

$$\phi_\Delta^{\text{clamp}}(x) = x + \texttt{stopgrad}\big(\text{clamp}(x, -\xi, \xi) - x\big). \tag{13}$$

We empirically verify the importance of both the Huber kernel and clamping in our experiments.

## 2.4 FINAL FPO++ OBJECTIVE

We now summarize the complete FPO++ algorithm by combining the training modifications described above. The key differences from vanilla FPO are:

1. Ratios are computed *per-sample* rather than per-action, increasing effective batch size.
2. We use the SPO (Xie et al., 2024) surrogate objective for negative advantages.
3. The CFM loss uses a Huber kernel with clamping to improve numerical stability.

Formally, for each action $a_t$ with advantage $\hat{A}_t$, we draw $N_{\text{mc}}$ Monte Carlo pairs $(\tau_i, \epsilon_i)$. We compute the robust CFM loss using each pair:

$$\ell_\theta^{\text{Huber}}(a_t, \tau_i, \epsilon_i; o_t). \tag{14}$$

The final FPO++ objective is then

$$\max_{\theta} \quad \mathbb{E}_{a_t \sim \pi_{\theta_{\text{old}}}(a_t|o_t)} \left[ \frac{1}{N_{\text{mc}}} \sum_{i=1}^{N_{\text{mc}}} \psi_{\text{ASPO}}\left( \hat{r}_{\text{FPO}}^{(i)}(\theta), \hat{A}_t \right) \right], \tag{15}$$

where the per-sample ratio is

$$\hat{r}_{\text{FPO}}^{(i)}(\theta) = \exp\Big( \phi_\Delta^{\text{clamp}}\big(\ell_{\theta_{\text{old}}}^{\text{Huber}}(a_t, \tau_i, \epsilon_i; o_t) - \ell_\theta^{\text{Huber}}(a_t, \tau_i, \epsilon_i; o_t)\big) \Big), \tag{16}$$

and the ASPO trust region objective is defined piecewise:

$$\psi_{\text{ASPO}}(r, \hat{A}_t) = \begin{cases} \min\Big( r\,\hat{A}_t,\; \text{clip}\big(r, 1 - \varepsilon^{\text{clip}}, 1 + \varepsilon^{\text{clip}}\big)\,\hat{A}_t \Big), & \hat{A}_t > 0, \\ r\,\hat{A}_t - \dfrac{|\hat{A}_t|}{2\,\varepsilon^{\text{clip}}}\,(r - 1)^2, & \hat{A}_t \leq 0. \end{cases} \tag{17}$$

We empirically verify this formulation in our experiments.

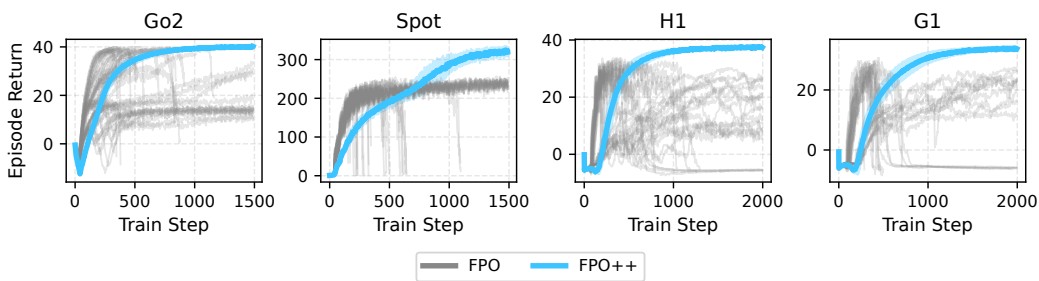

Figure 2: **FPO vs FPO++ stability.** We compare episode returns from FPO++ training with FPO returns over many different hyperparameter choices. Algorithmic changes in FPO++ resolve training performance and stability problems that cannot be solved by tuning FPO hyperparameters.

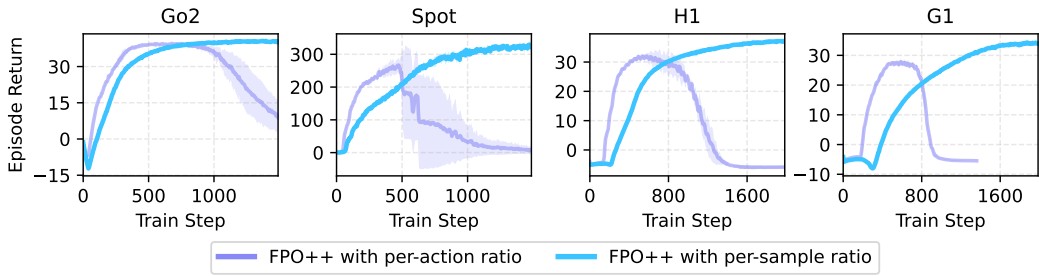

Figure 3: **Ablation on ratio approximation.** FPO++ uses a per-sample ratio approximation, which results in more stable training than FPO's per-action ratio.

**Relation to architectural stabilization methods.** One natural question is whether the instabilities described in Section 2.2 could be addressed through architectural strategies (Nauman et al., 2024; Lee et al., 2024) rather than algorithmic changes. These architecture changes are designed for instability when scaling to higher-capacity networks; in contrast, we consider stability problems for smaller networks used in real-time robot control. Hyperparameters and network architecture details can be found in the Appendix. We also found that hyperparameter tuning and standard PPO implementation details (running observation normalization, gradient clipping, weight decay, etc) do not prevent the collapse or post-peak decay of FPO. In contrast, FPO++ is stable with a broad range of hyperparameter choices.

## 3 EXPERIMENTS

The goal of our experiments is to validate and evaluate FPO++ on real robotics problems. To accomplish this, we train policies for both legged locomotion and humanoid motion tracking.

We structure our experiments as follows. **(i)** We begin by evaluating policy learning from scratch using simulated locomotion tasks, on both quadrupedal and bipedal robots (Section 3.1). **(ii)** We ablate design decisions, including the asymmetric trust region objective and per-sample ratio. **(iii)** We analyze the policy distribution that FPO++ learns. **(iv)** Finally, we show that FPO++ trained in simulation can be zero-shot deployed to real humanoid robots.

### 3.1 LOCOMOTION BENCHMARKING

To evaluate the characteristics of FPO++, we begin by training FPO++, FPO, and Gaussian PPO policies using the standard IsaacLab (Mittal et al., 2023) velocity-conditioned robot locomotion environments. We include results for four simulated robots: Unitree Go2, Boston Dynamics Spot, Unitree H1, and Unitree G1. Results and analysis are discussed below; hyperparameter sweeping procedures and implementation details are documented in the appendix.

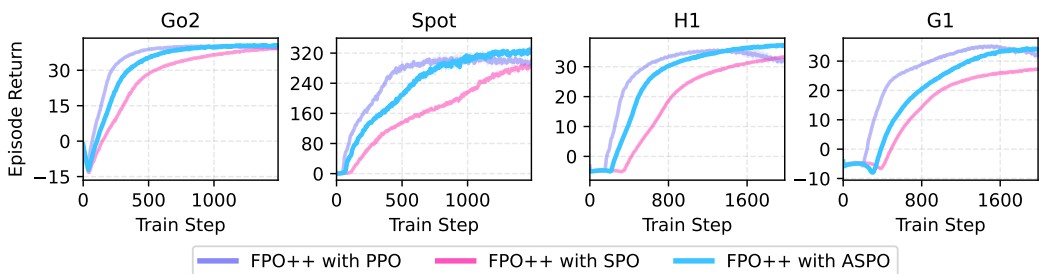

Figure 4: **Ablation on trust region objective.** We compare replacing FPO++'s asymmetric clipping objective with standard PPO-style clipping and an SPO (Xie et al., 2024) trust region. FPO++ balances training speed and stability, while converging to the highest reward across tasks.

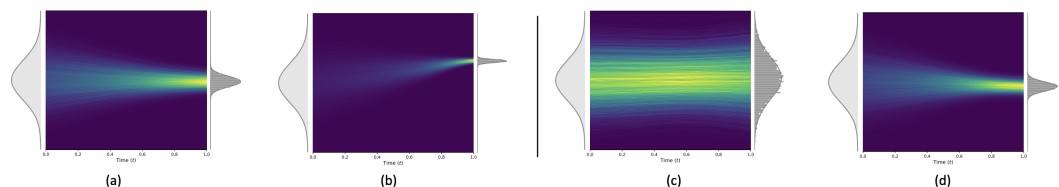

(a)                    (b)                    (c)                    (d)

Figure 5: **ASPO preserves entropy.** We visualize the flow field density for a single action dimension of H1 humanoid locomotion. An FPO++ policy trained with a standard PPO trust region is shown at its peak reward (a) and after its performance has started to degrade (b). The narrowing of the distribution in (b) illustrates an entropy collapse, leading to instability. In contrast, FPO++ with an ASPO objective is shown at a checkpoint with a reward level similar to the baseline's peak (c) and at its final, higher-reward converged state (d). ASPO maintains a wider, more exploratory distribution; it better preserves entropy and prevents policy collapse.

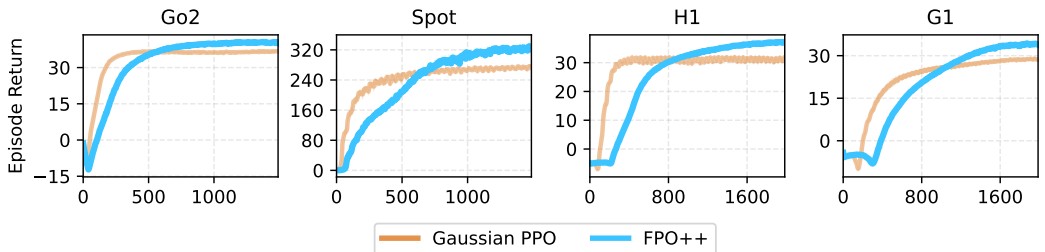

Figure 6: **Comparison against Gaussian PPO.** We compare FPO++ training curves against PPO; FPO++ compares favorably.

**FPO++ trains stably.** Figure 2 reports training curves of FPO++ averaged over 5 seeds, compared with FPO runs that sweep over hyperparameters like learning rate $\in \{0.00001, 0.0001, 0.0003\}$, clip parameter $\in \{0.04, 0.05, 0.06\}$, and Monte Carlo samples $\in \{8, 16, 32\}$. We found that standard FPO struggles significantly in these tasks, even after significant tuning of these parameters and adding incorporating details like adaptive learning rates, gradient clipping, and running observation normalization. In contrast, FPO++ stably solves all locomotion tasks.

**Ratio approximation ablation.** A primary difference between FPO++ and FPO is an updated ratio approximation. We evaluate the effect of this in Figure 3. The original FPO approximation converges more slowly and often leads to unstable training, while the FPO++ variant, although introducing a small bias, results in decreased variance in gradient estimation that improves the training stability. The reduced variance is especially important in the later stage of training, when the policy has low entropy and high gradient variance is more likely to cause collapse, as the baseline shows.

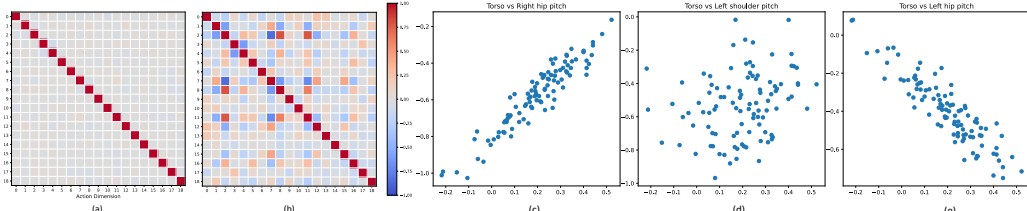

Figure 7: **The FPO++ objective results in meaningful action distributions.** We sample many actions conditioned on the same observation from a policy learned for H1 humanoid locomotion. The policy at initialization is visualized in (a): it exhibits uncorrelated actions, similar to a diagonal Gaussian policy. In contrast, the converged policy (b) after FPO++ training learns significant off-diagonal correlations, which verifies its ability to capture more action dimension dependencies than is possible with standard diagonal Gaussian policies. Correlations during locomotion are interpretable: the right hip (c) correlates positively with the torso, the left shoulder (d) is uncorrelated, and the left hip (e) correlates negatively.

**Trust region ablation.** To evaluate the effectiveness of different trust region objectives, we compare their learning curves in Figure 4. Replacing the asymmetric clipping objective with PPO-style clipping results in slightly faster early-stage learning but leads to instability in later training. On the other hand, adopting the objective from SPO (Xie et al., 2024) improves stability but significantly slows down progress. In contrast, FPO++, which employs an asymmetric trust region, achieves both stability and efficiency throughout training.

To better understand the differences in training performance, we examine how policy distribution progresses during training, as shown in (Figure 5a–d). We see that vanilla FPO with the standard PPO trust region produces highly centered action distributions, leading to entropy collapsing and performance deteriorating over time (Figure 5a–b). In contrast, FPO++ maintains a broader, more exploratory distribution even after convergence (Figure 5c–d), contributing to stable and robust performance. This behavior is driven by the asymmetric trust region, which implicitly regularizes entropy and prevents collapse. These dynamics align with the learning curves in Figure 4, where FPO++ outperforms PPO and vanilla FPO by preserving exploration and ensuring training stability. Notably, despite these differences in training stability, both the vanilla FPO and FPO++ policies learn a seemingly Gaussian exploration strategy.

**Comparison against Gaussian PPO.** We compare FPO++ training curves against Gaussian PPO training curves in Figure 6. While outperforming Gaussian PPO is not the purpose of this work, we nonetheless find that FPO++ achieve competitive training returns. We note that one limitation of FPO++, however, is wall-clock time: FPO++ is slower to train than Gaussian PPO because backprop is required through each CFM Monte Carlo sample. The runtime hit for reported experiments is typically around 20%: for G1 locomotion on an L40S GPU, our Gaussian PPO baseline reaches a return of 25 in 19 minutes. FPO++ experiments required 23 minutes to reach the same return. Future improvements to FPO++ may explore adaptive sampling techniques to accelerate training without sacrificing policy performance.

**Analyzing FPO++ policy distributions.** A key advantage of flow-based models is the ability to represent complex distributions. To understand what FPO++ learns in practice, we analyze the resulting policy distribution in the locomotion task by taking multiple samples at fixed observations and visualizing correlations between actions dimensions. We find interpretable correlations, which are visualized and discussed in Figure 7.

## 3.2 SIM-TO-REAL WITH FPO++

We validate the real-world capabilities of FPO++ by deploying policies on two distinct hardware platforms: a Booster Robotics T1 humanoid for a velocity-conditioned locomotion task and a Uni-

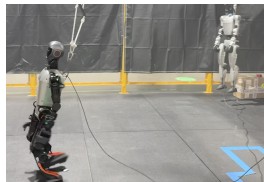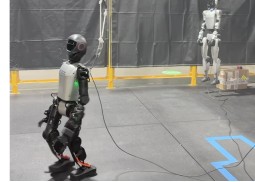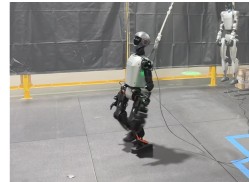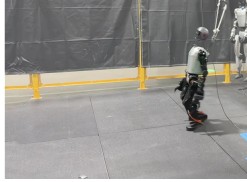

Figure 8: **FPO++ policy deployment on Booster T1 robot.** We deploy a joystick-conditioned locomotion policy to the Booster T1 humanoid robot.

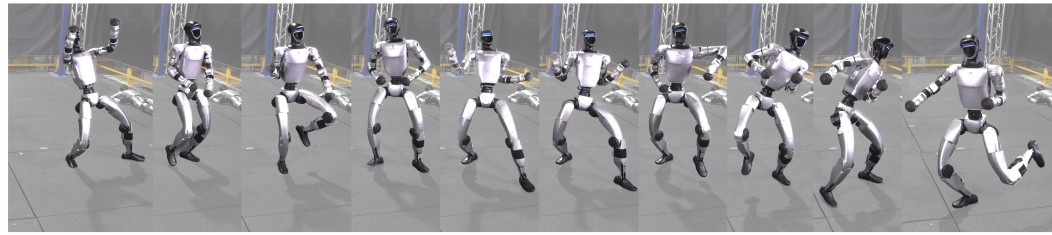

(a) **Dynamic motion tracking:** The FPO++ policy reliably tracks long (1.5-minute) and highly dynamic dancing motions from the LAFAN dataset entirely onboard.

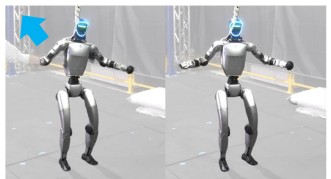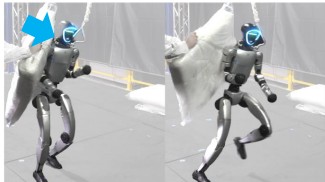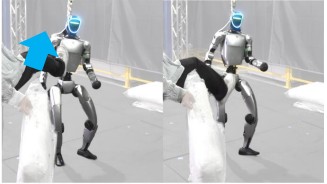

(b) **Robustness to Perturbations:** The FPO++ policy exhibits strong robustness against external forces and pushes, maintaining balance and continuing to track the reference motion despite significant disturbances.

Figure 9: **FPO++ motion tracking policy deployed on a Unitree G1 robot.** We trained FPO++ for BeyondMimic (Liao et al., 2025)-style motion tracking task and deployed it to a physical G1 robot in zero-shot.

|  | FPO++ | PPO |
|---|---|---|
| Return | $38.4 \pm 0.6$ | $37.4 \pm 1.1$ |
| Episode Length | $452.5 \pm 5.2$ | $475.4 \pm 9.6$ |

Table 1: **Motion tracking metrics.** We report training return and episode length statistics from FPO++ and Gaussian PPO, for BeyondMimic (Liao et al., 2025)-based motion tracking.

tree G1 humanoid for motion tracking. Policies are trained in simulation and transferred zero-shot to physical robots in realworld. Videos can be found on our project webpage [2].

**Setup.** The T1 locomotion policy was trained using a modified version of the Humanoid-Verse (LeCAR-Lab, 2023) framework, which uses IsaacGym (Makoviychuk et al., 2021) for simulation. The G1 motion tracking policy was trained by adapting the BeyondMimic (Liao et al., 2025) codebase, which is built on IsaacLab framework (Mittal et al., 2023); we replaced its PPO objective and MLP actor with the FPO++ objective and a flow model, respectively. We used LAFAN motion capture data (Harvey et al., 2020)[3] retargeted to Unitree G1 to train a policy.

**Evaluation.** FPO++ is the first method to demonstrate successful zero-shot sim-to-real transfer for a humanoid robot using a flow matching policy trained entirely from scratch with on-policy

---

[2]Sim-to-real humanoid videos: https://fpocontrol.github.io/

[3]This dataset was used solely for research purposes.

reinforcement learning. Here, we focus on the qualitative evaluation of the sim-to-real performance of FPO++ across the two hardware platforms following prior work, since quantitative comparison of sim-to-real methods remains challenging due to the variability of physical conditions and task setups.

First, FPO++ learns robust gaits for Booster T1 locomotion (Figure 8). Second, the FPO++ policy reliably tracks long, dynamic motion sequences while remaining robust to external perturbations (Figure 9). Although the showcased motion-tracking result is trained on a single sequence, as in BeyondMimic, where the policy quickly saturates the reward, we also evaluate training performance in simulation by training FPO++ and a Gaussian PPO baseline to track the entire LAFAN dataset. Tracking the full corpus is challenging, but FPO++ achieves performance comparable to Gaussian PPO (Table 1).

**Inference strategy.** We outline two practical inference-time considerations for sim-to-real deployment of FPO++, detailed below.

*Sampling step count.* Flow policies trained by FPO and FPO++ are compatible with any choice of sampler schedule. We found that a small number (4 or 8) of flow sampling steps could be used for the ODE integration to generate actions with acceptable latency.

*Smoother test-time actions.* Reinforcement learning with policy gradient methods requires stochastic policies during training: the policy outputs a distribution from which actions are sampled, enabling exploration. At test-time, however, it is standard to switch to a deterministic inference strategy. For Gaussian policies, this typically corresponds to taking the mean action. We found a similar strategy beneficial for flow policies. During training, initial noises are drawn from $\epsilon \sim \mathcal{N}(0, I)$; at test-time, we instead initialize flow integration from $\epsilon = \vec{0}$. We found that this inference procedure produces smoother, more stable actions that qualitatively improves performance across tasks.

## 4 RELATED WORK

**Flow-based Policies for Robot Control.** Recently, denoising diffusion models (Chi et al., 2024b; Ankile et al., 2024; Reuss et al., 2023) have shown success in robot manipulation tasks, where policies are trained with human demonstrations (Mandlekar et al., 2021) through supervised learning. In these approaches the policy is modeled using diffusion or flow-based models, an expressive class of generative models capable of learning multi-modal distributions. $\pi_0$ (Black et al.) adopts a diffusion model for large-scale VLA-based manipulation, and $\pi_0.5$ (Black et al., 2024) later uses a flow matching model. Recently, Streaming Flow Policy (Jiang et al., 2025) proposed to turn the flow process over action trajectories into a streaming flow in action space, enabling faster and reactive policy. In this paper, we treat diffusion and flow policies as a single family of iterative generative policies. Going beyond manipulation, Diffuse-CLoC (Huang et al., 2025) proposes a guided diffusion policy for physics-based, whole-body control via supervised learning, in which a single diffusion model jointly generates *(state, action)* horizons to enable steerable, look-ahead control. BeyondMimic (Liao et al., 2025) extends this line to a real humanoid by performing an offline distillation from multiple motion tracking expert policies to a single diffusion policy. All of these existing flow-based and diffusion policies rely on human demonstrations or distillation from expert controllers. In contrast, we propose FPO++, an on-policy reinforcement learning algorithm that is trains flow policies from scratch in a reinforcement learning setting.

**Online Reinforcement Learning for Flow-based Policies.** In online RL for robot control, DPPO (Ren et al., 2024) treats the denoising process as a Markov Decision Process (MDP). Like DDPO (Black et al., 2023), this uses noise injected via a stochastic sampler to chain Gaussian likelihoods. This is compatible with standard policy gradient techniques, but increases the task horizon and ties the underlying diffusion model to a specific sampler choice. ReinFlow (Zhang et al., 2025b) applies a similar approach to flow policies by injecting learnable noise into the deterministic flow path, converting it into a discrete-time Markov process for likelihood computation. NCDPO (Yang et al., 2025) reformulates the diffusion process as a noise-conditioned deterministic policy and back-propagates gradients through all diffusion timesteps, rather than treating diffusion timesteps as an MDP to not increase a task horizon and make credit assignment stable. GenPO (Ding et al., 2025) leverages exact diffusion inversion to construct invertible action mappings and introduces a "doubled dummy action" mechanism that enables invertibility via alternating updates, yielding tractable

log-likelihoods. In contrast, FPO (McAllister et al., 2025) offers a simpler and more direct alternative. Specifically, FPO does not extend the task horizon, require backpropagation through the generative steps, or depend on invertible architectures, and it is agnostic to the choice of sampling method. This streamlined design makes it particularly effective for challenging control problems. FPO++ is the first method to demonstrate successful zero-shot sim-to-real transfer for a humanoid robot using a flow-matching policy trained from scratch with on-policy RL.

## 5 CONCLUSION

In this paper, we introduce a practical training recipe for training flow policies for real robotic systems. We demonstrate that FPO++ can effectively train policies for complex legged robots, which can also successfully transfer from simulation to real hardware. We hope this existence proof helps spur further research in this area. To establish reliable training foundations, we intentionally keep the flow policy architecture simple, focusing on algorithmic stability rather than architectural innovations. Future work may explore policies conditioned on more observation and state history, training flow policies that predict multiple future actions (action chunking), and fine-tuning of behavior-cloned policies. We hope our findings and open-source implementation will facilitate progress in these endeavors.

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

# Appendix of "Flow Policy Gradients for Legged Robots"

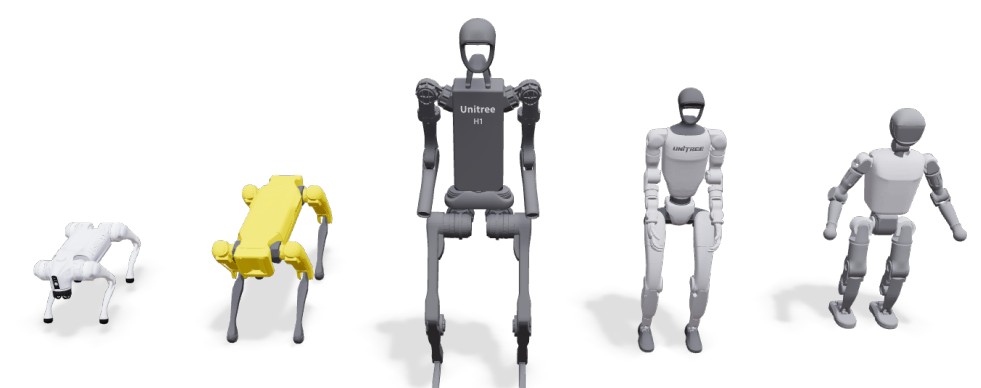

Figure A.1: **Robots used for experiments.** We train policies for the Go2, Spot, H1, and G1 robots in simulation. We deploy policies to physical G1 and Booster T1 robots.

## A   FURTHER RELATED WORK

**Policy gradients for robot control.**   Policy gradient methods, such as Trust Region Policy Optimization (TRPO) (Schulman et al., 2015a) and Proximal Policy Optimization (PPO) (Schulman et al., 2017) have demonstrated remarkable success across diverse robot control tasks. They enable quadrupeds to recover from falls on challenging terrain (Lee et al., 2019), achieve locomotion from pure proprioception without exteroceptive sensing (Lee et al., 2020), and perform agile, high-speed maneuvers (Hwangbo et al., 2019). More recently, Liu et al. (2025) have shown PPO can scale to generalist locomotion across diverse legged morphologies.Beyond locomotion, policy gradients drive example-guided imitation of character skills (Peng et al., 2018), contextual humanoid behaviors such as terrain traversal and stair climbing from monocular video (Allshire et al., 2025), and hierarchical humanoid planning for tasks like table tennis rallies (Su et al., 2025). In manipulation, they have enabled solving a Rubik's cube with a humanoid hand via automatic domain randomization (Akkaya et al., 2019) and vision-based dexterous in-hand manipulation under large-scale randomization (Handa et al., 2022). Despite these successes, PPO variants in continuous control are designed for Gaussian policies, whose modeling capabilities are fundamentally limited. In this work, we introduce an RL training recipe for flow-based policies—policies that leverage flow models, an expressive class of generative models—within a PPO-like framework that can be trained stably on robot tasks.

**Flow-based offline RL.**   A large body of offline RL work has explored using flow and diffusion-based generative models to represent policies. One common strategy is advantage weighted regression (AWR) Peng et al. (2019), where the generative model is trained with advantage-weighted updates (Kang et al., 2024; Lu et al., 2023; Ding et al., 2024; Zhang et al., 2025a). Another line of work optimizes a Q-learning objective jointly with a generative-model loss (Wang et al., 2022; He et al., 2023; Ding & Jin, 2023; Zhang et al., 2024; Ada et al., 2024), enabling value-based training while regularizing the policy through diffusion or flow matching. Maximum-entropy approaches such as DIME Celik et al. (2025) extend this direction by integrating diffusion policies with entropy-regularized RL, further improving robustness and sample quality in offline settings.

FQL Park et al. (2025) takes a different approach by training a one-step flow policy without back-propagation through time (BPTT), avoiding instability issues that arise in iterative policy improvement. Our method, based on FPO (McAllister et al., 2025), is conceptually closest to AWR in its use of an advantage-weighted flow-matching loss. However, FPO is fundamentally distinct: it is an on-policy algorithm that learns directly from environment interaction rather than a static offline dataset, and it avoids BPTT without requiring distillation from a behavior policy as in FQL.

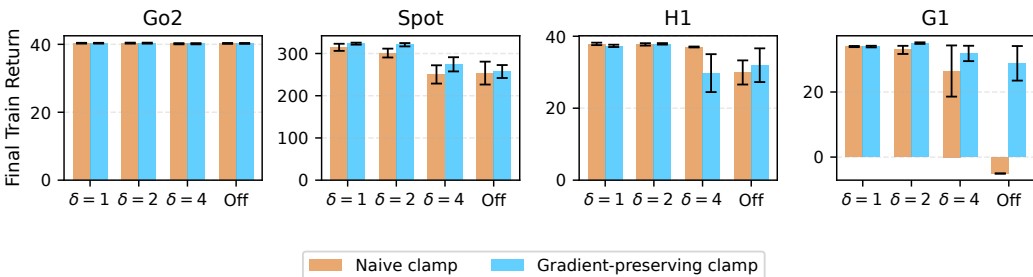

Figure A.2: **Huber loss and gradient-preserving clamping improve training.** We present training returns for FPO++ runs after sweeping different Huber parameters ($\delta = \{1, 2, 4, \text{Off}\}$) and both naive clamping and gradient-preserving clamping (Equation 13). We observe that the same Huber $\delta = 2$ and gradient-preserving clamping can be used to improve results across environments. Mean and standard error are computed over 5 seeds.

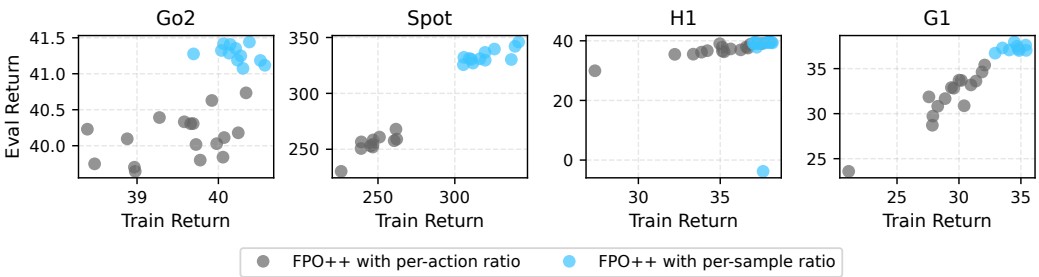

Figure A.3: **Per-sample ratio improves training and evaluation returns.** We sweep the clipping parameter over $\{0.04, 0.05, 0.06\}$ and observe that FPO++ with per-sample ratios consistently achieves higher train and eval returns than the per-action ratio formulation.

## B  ADDITIONAL EXPERIMENTS FOR REBUTTAL

In response to reviewer feedback, we conducted several additional experiments to validate the design choices in FPO++ and to provide further empirical evidence for the mechanisms discussed in the paper. Figure A.2 presents a sweep over Huber parameters and clamping strategies, showing that $\delta = 2$ together with gradient-preserving clamping improves training stability across all environments. Figure A.3 demonstrates that using per-sample ratios consistently improves both training and evaluation returns over a range of clipping thresholds. Figure A.4 compares PPO, SPO, and ASPO trust regions and shows that ASPO provides higher and more stable performance across hyperparameter settings. Finally, Figure A.5 provides direct evidence for reduced gradient variance by showing increased cosine similarity of per-update gradients when using ASPO and per-sample ratios. Together, these results support the algorithmic modifications introduced in FPO++ and address core concerns raised by reviewers.

## C  EXPERIMENT DETAILS

### C.1  LOCOMOTION BENCHMARKING

#### C.1.1  HYPERPARAMETERS

All policies for the locomotion benchmarking experiments utilize 3-layer Multi-Layer Perceptrons (MLPs). Specifically, the actor network employs 256 hidden units, and the critic network employs 768 hidden units per layer. Additional hyperparameters can be found in Table A.1.

Quadruped policies are trained for 1500 steps, and humanoid policies are trained for 2000 steps. For the Flow Policy Optimization (FPO) algorithm, the clipping parameters are set to 0.05 for quadrupeds and 0.03 for humanoids. For the Proximal Policy Optimization (PPO) algorithm, we

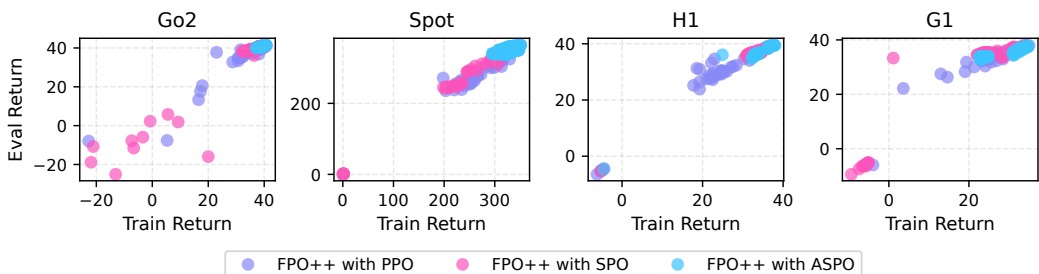

Figure A.4: **ASPO improves training and evaluation returns.** We sweep clipping and learning-rate settings for FPO++ using PPO, SPO, and ASPO trust regions, and plot the resulting train and eval returns. ASPO consistently yields higher and more stable performance.

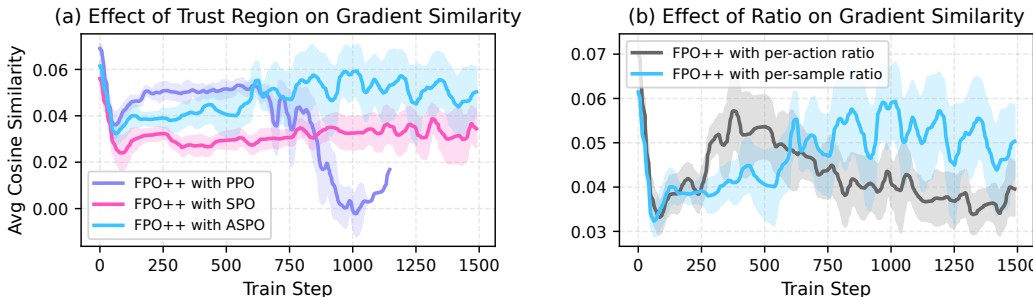

Figure A.5: **ASPO and per-sample ratios reduce empirical gradient variance.** We measure gradient variance by adapting the cosine similarity metric employed by (Ilyas et al., 2020): for each policy update, we measure the average cosine similarity between individual gradients and the average of gradients within that policy update. We observe that ASPO and the per-sample ratio result in higher cosine similarity between gradients computed within each policy update; this provides evidence of lower gradient variance. Averages are reported over 5 seeds; the PPO curve terminates early because of one of these seeds crashes from numerical instability..

adopt the default configurations provided by the `rsl_rl` library, with additional sweeps performed over the clipping parameters in the set $\{0.1, 0.15, 0.2, 0.25\}$.

### C.1.2 ENVIRONMENTS AND REWARDS

All locomotion benchmarking experiments use the standard IsaacLab velocity-conditioned locomotion environments. We use the default weight for each reward term. The reward terms are as follows:

- **Linear velocity tracking**: The agent receives an exponential reward for matching the commanded linear velocity in the horizontal plane.
- **Angular velocity tracking**: The agent receives an exponential reward for matching the commanded yaw rate.
- **Feet air time**: The agent is rewarded for keeping feet in the air for a sufficient duration during each step.
- **Gait synchronization**: The agent is rewarded for synchronizing diagonal foot pairs to encourage a trotting gait.
- **Foot clearance**: The agent is rewarded for achieving a target foot clearance height during the swing phase.
- **Vertical velocity**: The agent is penalized for vertical base velocity.
- **Body orientation**: The agent is penalized for deviations from a flat base orientation.
- **Angular velocity (xy)**: The agent is penalized for roll and pitch angular velocities.
- **Joint torques**: The agent is penalized for high joint torque magnitudes.

- **Joint accelerations**: The agent is penalized for large joint accelerations.

- **Action rate**: The agent is penalized for rapid changes in actions between consecutive timesteps.

- **Foot slip**: The agent is penalized for foot sliding while in contact with the ground.

- **Joint limits**: The agent is penalized for joint positions that exceed soft limits.

- **Undesired contacts**: The agent is penalized for undesired contacts on non-foot body parts.

For more details, we refer to the open-source IsaacLab (Mittal et al., 2023) repository.

## C.2 MOTION TRACKING

Our motion tracking experiments train a policy for the Unitree G1 robot, which has 29 degrees of freedom (DoF). The control frequency is set to 50 Hz (with a simulation timestep $\Delta t = 0.005$ s and decimation of 4). The reward design, observation space, and termination conditions follow BeyondMimic (Liao et al., 2025)'s settings. The following sections detail the specific hyperparameters used.

### C.2.1 DOMAIN RANDOMIZATION

Domain Randomization (DR) is applied to improve the policy's ability to transfer to the real world:

- Physics Material: Static friction (Uniform $[0.3, 1.6]$), Dynamic friction (Uniform $[0.3, 1.2]$), and Restitution (Uniform $[0.0, 0.5]$) are sampled at startup.

- Joint Defaults: Default joint angles are uniformly offset by $\mathcal{U}(-0.01, 0.01)$ rad at startup.

- Center of Mass (COM): The torso COM is offset uniformly in $(x, y, z)$ at startup.

- External Forces: Pushes are applied at uniform intervals (2.0-3.0 s), with randomized Linear velocity $\mathcal{U}([(-0.5, 0.5), (-0.5, 0.5), (-0.2, 0.2)]$ m/s and Angular velocity $\mathcal{U}([(-0.52, 0.52), (-0.52, 0.52), (-0.78, 0.78)]$ rad/s).

- Actuator Command Delay: Actuator latency is simulated by applying a random delay of 0 to 2 simulation steps (corresponding to 0 ms to 10 ms at $\Delta t = 0.005$ s) to the control commands (joint positions, velocities, efforts) at each environment reset to improve robustness and sim-to-real transfer.

- Motion Initialization: Root position, orientation, and joint angle offsets are applied uniformly at episode reset.

| Hyperparameter category | Used value | Sweep range / Notes |
|---|---|---|
| *FPO++* | | |
| Flow integration steps | 32 | $\{8, 16, 32\}$ |
| Network output | $u$ | $\{u, x_0\}$, $x_0$ denotes data |
| Samples per action | 16 | $\{8, 16, 32\}$ |
| *Training* | | |
| Learning rate | $1 \times 10^{-4}$ | $1 \times 10^{-5}, 1 \times 10^{-4}, 3 \times 10^{-4}$ |
| Weight decay / Adam betas | $1 \times 10^{-4}$ / $(0.9, 0.95)$ | AdamW parameters |
| Clip parameter | 0.05 | $\{0.03, 0.04, 0.05, 0.06\}$ |
| Discount factor ($\gamma$) | 0.99 | |
| GAE lambda ($\lambda$) | 0.95 | |
| Learning epochs | 16 for Go2, others 32 | |
| Minibatches per update | 4 | |
| Running observation normalization | Yes | |

Table A.1: **FPO++ and training hyperparameters used for locomotion.** Reported sweeps are performed by modifying this standard configuration.

| Hyperparameter category | Used value | Sweep range / Notes |
|---|---|---|
| *FPO++* | | |
| Flow integration steps | 50 | $\{10, 50\}$ |
| Network output | $u$ | $\{u, x_0\}$, $x_0$ denotes data |
| Samples per action | 16 | $\{8, 16, 32\}$ |
| *Training* | | |
| Learning rate | $3 \times 10^{-4}$ | |
| Weight decay / Adam betas | $1 \times 10^{-4}$ / $(0.9, 0.95)$ | AdamW parameters |
| Clip parameter | 0.01 | $\{0.01, 0.05, 0.1\}$ |
| Discount factor ($\gamma$) | 0.99 | |
| GAE lambda ($\lambda$) | 0.95 | |
| Learning epochs | 5 | |
| Minibatches per update | 4 | |
| Running observation normalization | Yes | |

Table A.2: **FPO++ and training hyperparameters used for motion tracking.**

### C.2.2 TRAINING HYPERPARAMETERS

Training is conducted across 4096 parallel environments, with a rollout length of 96 steps per environment. Both the actor and critic networks use 3-layer policy MLPs for with hidden units of (1024, 512, 256). We will release the code for the further details. Additional hyperparameters can be found in Table A.2.

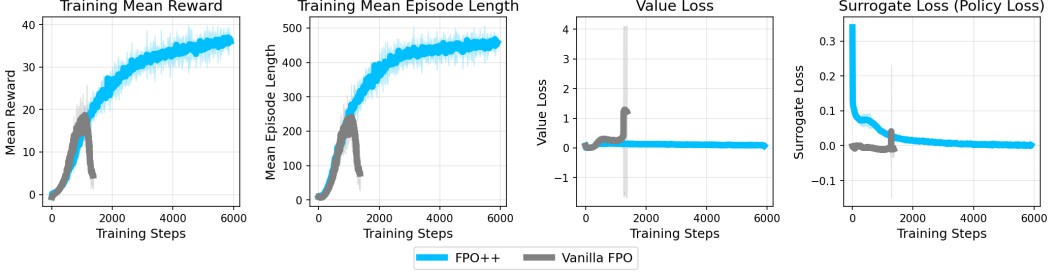

Figure A.6: **Motion tracking training curves.** FPO fails in this challenging, domain-randomized task, while FPO++ trains stably. Hyperparameters are shared between both runs.

### C.3 COMPARISON WITH VANILLA FPO

Figure A.6 shows the training curves for the motion-tracking policy that we deploy on the real G1 robot (Figure 9). As the plots illustrate, vanilla FPO initially learns but quickly collapses: the mean reward and episode length peak early and then deteriorate, while both the value loss and surrogate loss exhibit large spikes, indicating numerical instability. In contrast, FPO++ maintains stable value and policy losses throughout, continues improving monotonically, and successfully converges to the high-return policy used for real-world deployment. These curves provide direct evidence for the core motivation of our work: vanilla FPO is unstable on realistic, high-DoF robot control tasks, whereas the algorithmic components of FPO++ prevent collapse and enable successful sim-to-real transfer. We will release the full training code and configuration for reproducibility.

## D FULL ACTION SPACE FLOW FIELD

In addition to the qualitative analysis presented in Figures 4 and 5 of the main text, we provide visualizations of the flow fields across the full action space (19 DoF) for the H1 humanoid's *velocity commanded locomotion task*. These supplementary figures illustrate the impact of the trust region

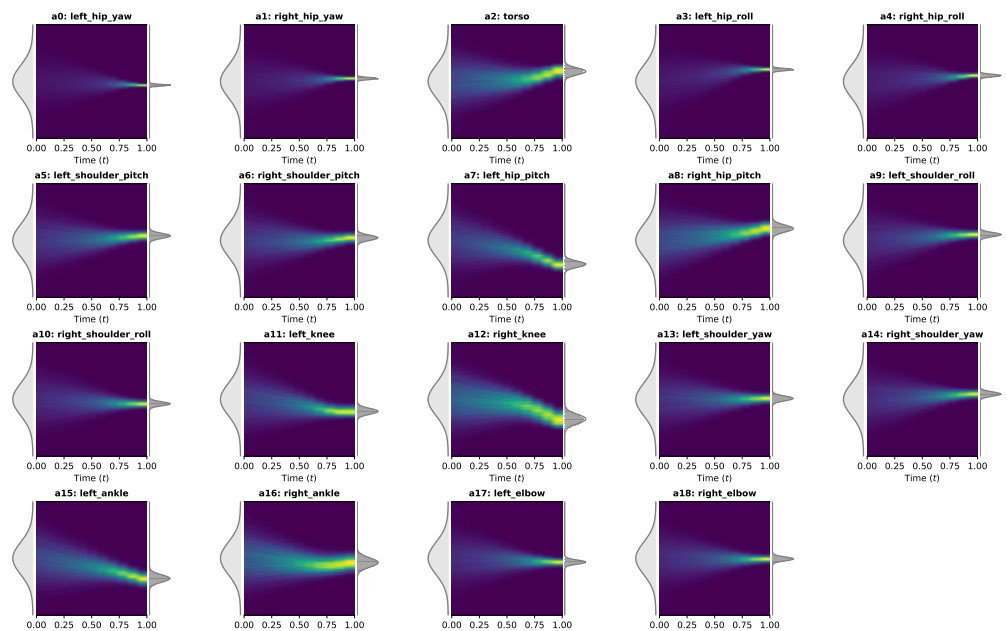

Figure A.7: **FPO++ with PPO clipping at peak performance**. Policy flow field density for the Unitree H1 humanoid trained using the standard PPO trust region, captured at the point of maximum average return. The flow is still relatively broad, indicating adequate exploration.

objective on the policy distribution at different stages of training. Specifically, the FPO++ policy trained with the standard PPO objective exhibits a clear narrowing of the action distribution (entropy collapse) as performance degrades, evident when comparing the field at its reward peak (Figure A.7) to the field during collapse (Figure A.8). In contrast, the policy trained with the asymmetric objective (ASPO) successfully maintains a broader, more exploratory distribution, consistently preserving entropy from an intermediate checkpoint (Figure A.9) to its final, converged state (Figure A.10). This confirms that the asymmetric objective effectively prevents the policy collapse shown in the main text, irrespective of the action dimension (joint type). For visualization, we plotted the transporting trajectories of the prior Gaussian noise to the action space by sampling 10,000 noises per state and using 8 discretized Euler integration steps.

## E MANIPULATION

We validate that the contributions of FPO++ generalize by fine-tuning policies on the RoboMimic (Mandlekar et al., 2021) tasks (Figure A.11), implemented in the Robosuite framework (Zhu et al., 2020), which is built on MuJoCo (Todorov et al., 2012) and operated at a 20 Hz control frequency. We first train a flow-matching base policy using the RoboMimic behavior-cloning dataset, following the data preparation procedure of Ankile et al. (2025). We then fine-tune this policy using FPO++ and compare its performance against Vanilla FPO as well as two established on-policy flow-based optimization baselines: DPPO (Ren et al., 2024) and ReinFlow (Zhang et al., 2025b). These baselines are particularly relevant because they represent contemporary approaches for applying on-policy RL methods to expressive generative policies (diffusion or flow models) without relying on residual actors. Our experiments focus on two representative single-arm manipulation tasks, Can and Square (Figure A.11, which use a Franka arm with a parallel-jaw gripper and a 7-dimensional action space (6 DoF end-effector delta pose plus a 1 DoF gripper action). The tasks involve challenges from sparse rewards, where a reward is only given upon successful task completion. FPO++ converges to a higher evaluation success rate than all baselines, as shown in Figure A.12.

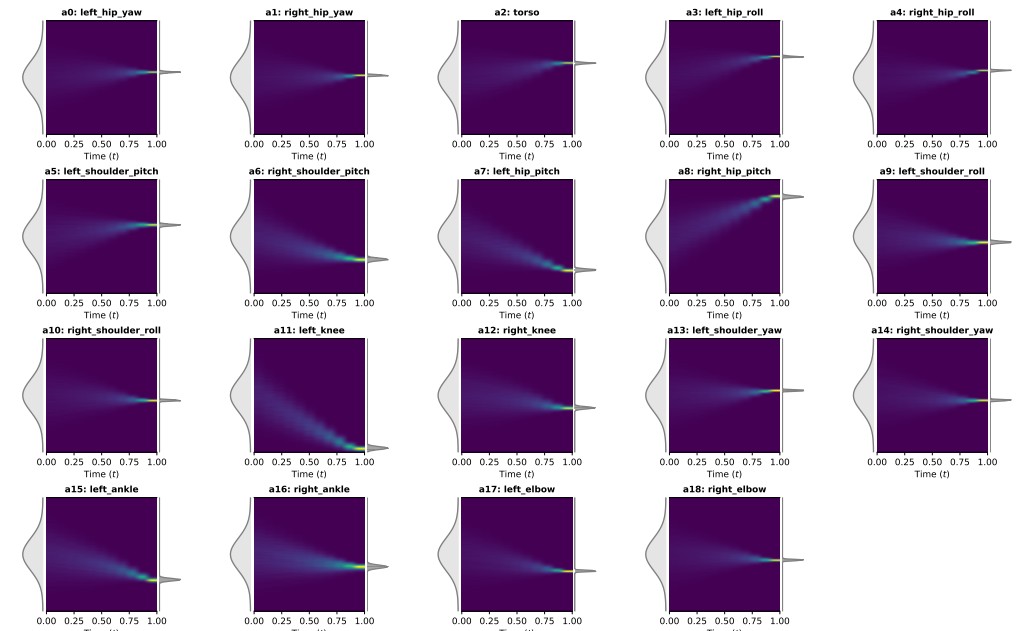

Figure A.8: **FPO++ with PPO clipping during policy collapse.** Policy flow field density for the Unitree H1 humanoid trained using the standard PPO trust region, captured as the average return begins to collapse. The action distribution has narrowed significantly for many joints, demonstrating the entropy collapse that leads to instability and performance degradation.

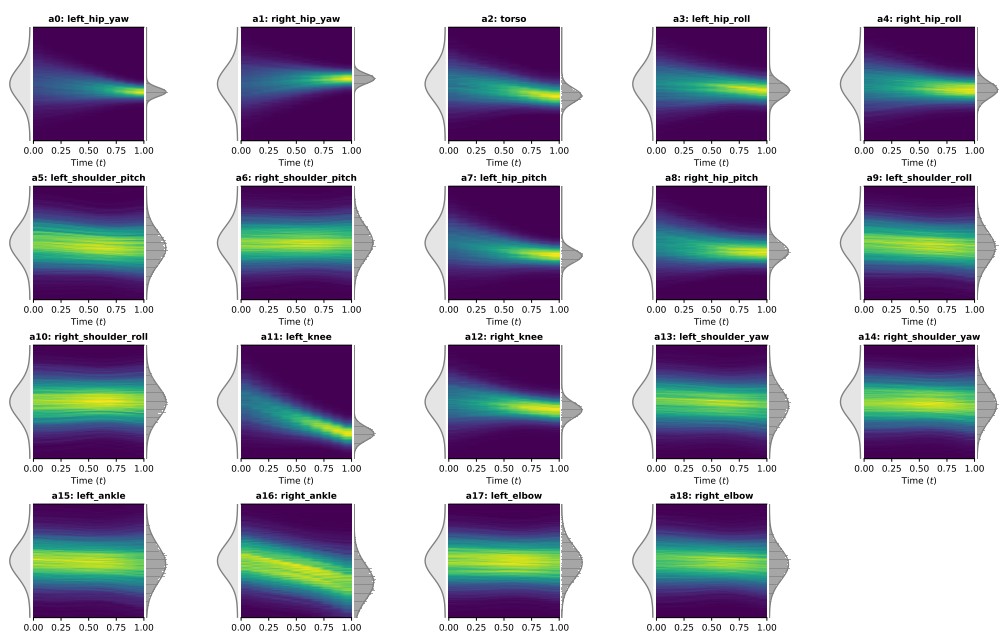

Figure A.9: **FPO++ with ASPO objective at intermediate training stage.** Policy flow field density for the Unitree H1 humanoid trained using the entropy-preserving ASPO objective, captured at an intermediate stage where the average return matches the peak average return of FPO++ with the standard PPO trust region. This distribution is already wider and more exploratory than the PPO-clipped policy at a similar or later time step, contributing to stable training.

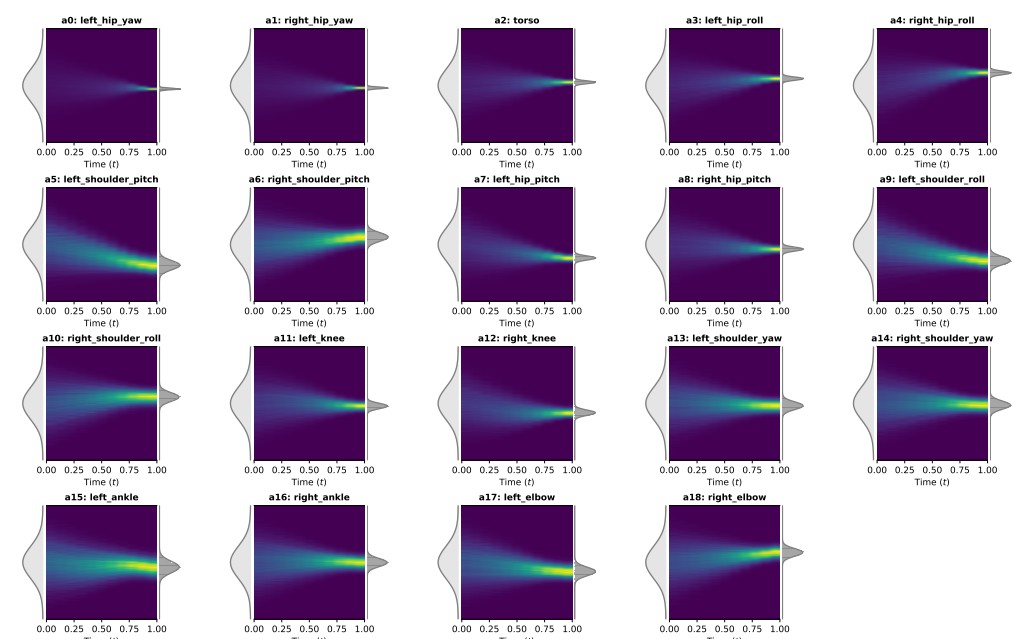

Figure A.10: **FPO++ with ASPO objective at converged state.** Policy flow field density for the Unitree H1 humanoid trained using the ASPO objective, captured at its final, high-reward converged state. The distribution remains wide and exploratory, confirming that ASPO effectively preserves entropy and prevents the collapse observed with PPO clipping.

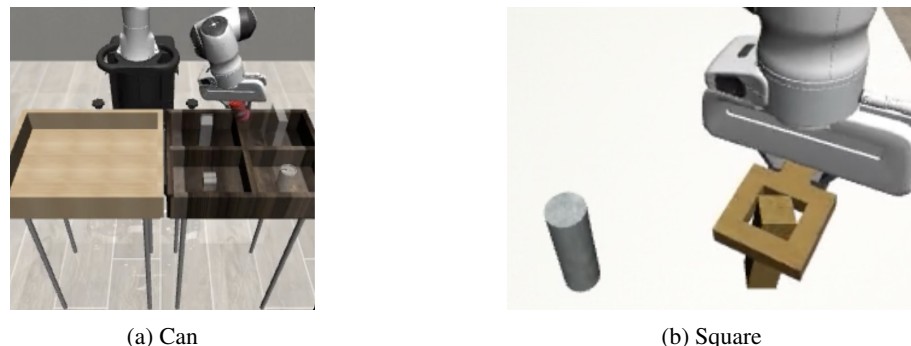

(a) Can                                    (b) Square

Figure A.11: **RoboMimic Can and Square manipulation tasks.** We evaluate FPO++ and baseline flow-based RL methods on two single-arm RoboMimic benchmarks implemented in Robosuite: (a) *Can*, where a Franka arm with a parallel-jaw gripper must grasp and move a can, and (b) *Square*, a more challenging task that requires precise object alignment and insertion using the same 7D action space (6 DoF end-effector delta pose plus 1 DoF gripper command).

**FPO++ implementation for manipulation.** To maintain architectural parity with DPPO and ReinFlow and to leverage short-horizon temporal correlations in the controls, all manipulation policies operate over *action chunks*. Following these baselines, the pre-trained base policies are structured to predict chunks of size 4. During fine-tuning, we apply the policy gradient objective at the chunk level: instead of computing a log-probability for each individual action, we approximate the log-probability of an action chunk by summing the corresponding CFM losses across the four actions in the chunk. This chunk-wise log-ratio is then used inside a PPO-style trust region objective, mirroring the training scheme of DPPO and ReinFlow while replacing their diffusion or flow likelihood with our CFM loss.

For all manipulation experiments, we use small learning rates for stable fine-tuning (actor learning rate $1 \times 10^{-5}$, critic learning rate $1 \times 10^{-4}$), a shared discount factor $\gamma = 0.99$, GAE parameter

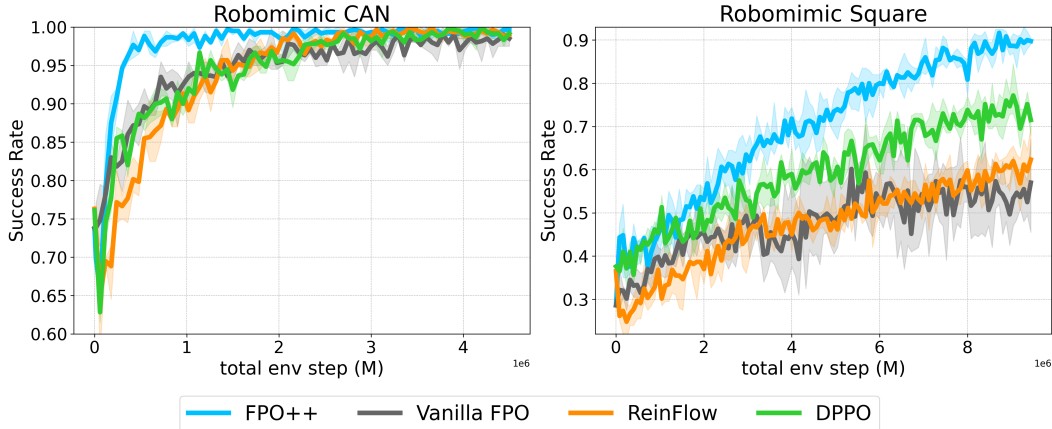

Figure A.12: **Manipulation comparison.** Fine-tuning success rates on the RoboMimic *Can* and *Square* tasks for flow-based RL methods. All methods start from pre-trained behavior-cloning base policies; FPO++ and Vanilla FPO share the same base policy, while DPPO and ReinFlow use architectures and implementation from their original papers. Please refer to the base policy setting written in the text for further details. FPO++ learns fastest and attains the highest final success on both tasks, while Vanilla FPO, DPPO, and ReinFlow remains stable but underperforms FPO, highlighting the benefit of the FPO++ training recipe for expressive manipulation policies.

$\lambda = 0.99$, and gradient clipping with max norm of 1. The vision encoder is kept frozen during fine-tuning, and we use zero-initialized flow integration for deterministic inference for FPO++. We will release the code for further details.

**Base policy setting.** The base policies' performance, reflected in the success rates at 0 total environment steps in Figure A.12, provides the anchor for fine-tuning. FPO++ and Vanilla FPO share the same base policy, while DPPO and ReinFlow utilize different pre-trained policies structured according to their original papers. The policies for DPPO and ReinFlow were trained and evaluated using 100 denoising/flow steps, whereas the base policy for FPO++ and Vanilla FPO was trained and evaluated with 10 flow steps. The base policy success rates (evaluated on 1,000 episodes) are: * **Can task**: FPO++/ Vanilla FPO (73.76%), ReinFlow (76.3%), and DPPO (76.1%). * **Square task**: DPPO (37.6%), ReinFlow (36.5%), and FPO++/ Vanilla FPO (28.62%).

**Analysis of fine-tuning performance** Figure A.12 plots the fine-tuning success rates (evaluated by collecting 200 episodes per timestep using 50 parallel environments). FPO++ achieves the highest final success rate in both the Can and Square tasks. In the Can task, FPO++ exhibits rapid initial learning, reaching high success significantly faster than all baselines. Notably, the Vanilla FPO variant performs robustly in this fine-tuning setup, achieving competitive results without the policy collapse or gradual decay seen in training from scratch on locomotion tasks (Figure 1 and Section 3.1). This suggests that initializing training with a stable pre-trained base policy provides sufficient regularization to mitigate the early-stage instability challenges inherent to vanilla FPO. Still, FPO++ achieves significant improvements over FPO-not only in learning speed, but also in its final performance on the Square task, which demands higher-precision control and is therefore more challenging. The successful application of FPO across manipulation, locomotion, and whole-body control highlights its general potential as a stable and expressive framework for diverse robotic tasks.

## F  ZERO-SAMPLING IN INFERENCE TIME

FPO++ generates actions by transporting an initial noise sample $\epsilon \sim \mathcal{N}(0, I)$ through a learned flow field. During training, the stochasticity of the learned flow field from different initial noise samples enable exploration and learning a rich action distributions. However, at inference time we have the freedom to choose how we initialize the flow integration. We found that a simple deterministic choice-*zero-sampling*, i.e., always setting $\epsilon = 0$ at test time-produces consistently higher returns and

lower variability across locomotion benchmarks, without changing the training procedure, as shown in Table A.3.

Zero-sampling is particularly important when deploying flow policies on real hardware under strict latency constraints. On the G1 motion-tracking task, the policy is trained with 50 flow integration steps but must be executed with as few as 5 steps on the robot to keep control latency acceptable. In this aggressively downsampled regime, standard random sampling becomes brittle: average performance degrades and variance increases, even though the underlying policy is the same. In contrast, zero-sampling remains robust to this step reduction, maintaining high mean rewards and substantially lower variance, as summarized in Table A.4. This robustness to reduced integration steps is crucial for sim-to-real deployment, where deterministic, low-latency inference is often a hard requirement.

| Robot | Random sampling | Zero-sampling |
|-------|-----------------|---------------|
| Spot | $331.3 \pm 2.4$ | $\mathbf{337.2 \pm 2.6}$ |
| Go2 | $40.6 \pm 0.0$ | $\mathbf{41.1 \pm 0.1}$ |
| H1 | $37.3 \pm 0.2$ | $\mathbf{38.4 \pm 0.1}$ |
| G1 | $34.4 \pm 0.5$ | $\mathbf{35.5 \pm 0.4}$ |

Table A.3: **Effect of inference-time action sampling on locomotion performance.** All policies are trained with stochastic sampling, but at test time we compare random-sampling versus zero-sampling. Zero-sampling consistently yields higher returns across all four robots.

| Sampling method | 5 steps | 50 steps |
|-----------------|---------|----------|
| Random sampling | $34.7 \pm 55.0$ | $38.4 \pm 26.6$ |
| Zero-sampling | $\mathbf{45.1 \pm 27.4}$ | $\mathbf{45.5 \pm 23.2}$ |

Table A.4: **Effect of zero-sampling on motion-tracking robustness.** Rewards are computed over 100 rollouts of a 2 min 30 s dancing sequence in IsaacSim, with domain randomization and strong push perturbations. The policy is trained with 50 flow steps; at test time we compare random sampling and zero-sampling under 5 and 50 flow steps.

