# OpenReview forum: "Flow Policy Gradients for Legged Robots"
_ICLR.cc/2026/Conference — Submitted to ICLR 2026_

### Official Review · Reviewer_zYv3 · 2025-10-30

**Soundness:** 3
**Presentation:** 3
**Contribution:** 2
**Rating:** 4
**Confidence:** 4

**Summary:**

This paper addresses the problem of learning control policies for high degree of freedom robots with complex dynamics, such as humanoid robots. The paper shows that a prior state of the art algorithm FPO suffered from problems of mode collapse, or gradual decay in performance. The authors propose a modified algorithm called FPO++ that contains four improvements to FPO, and show that FPO++ allows some complex behaviours to be learned.

**Strengths:**

The paper shows the limitations of FPO, and empirically delivers on the promised performance. The ideas are relatively clearly spelled out.

**Weaknesses:**

There isn't anything specifically wrong with this paper, but the contribution is somewhat limited, and there is not sufficient analysis of trade-offs presented by the four modifications. The four main changes from FPO to FPO++ are:

1. Ratios are computed per-sample rather than per-action, increasing effective
batch size.

The authors state that this modification decreases the gradient variance, but "it comes at the cost of bias.". The performance improvement is empirical, and while it may lead to more stable final policy returns, it would be important to perform analysis that says when this bias is problematic -- for example, if the samples are themselves high variance and have high dynamic range, then the gradient may actually vanish. Are there problems with this characteristic?

2. The SPO surrogate objective is used for negative advantages.

This objective "also reduces gradient variance" and empirically improves stability but again -- when is this problematic? Could the gradient vanish in this case too?

3. The CFM loss uses a Huber kernel with clamping to improve numerical
stability.

How did the Huber kernel get chosen? How easy is it to choose? Is there an interaction between the Huber kernel size and the trust region size?

4. At test-time, actions are sampled by initializing flow integration from
\epsilon = 0.

This seems to be a fairly minor step.

Given that these are the contributions, it would be important for the authors to give insight as to whether these changes in general are appropriate for most applications (should all applications of FPO use these modifications) or if they really are specific to legged locomotion in some way. This is especially important because the authors attribute the two main failures of FPO to numerical issues ("floating-point overflow" and "a small approximation error from Euler integration"). These would presumably be present in many applications.

The experimental results demonstrate generality across robots, but the paper is not precise about what the benchmarks are -- the cited paper is for Orbit which is a simulation environment paper, not a benchmark paper. Presumably the episode return in figures 2-6 are forward progress in some manner -- this is not stated.

The results in figures 7-9 are impressive, but do not necessarily add much to the overall paper in terms of convincing us that the authors have overcome some issues with numerical conditioning.

**Questions:**

Please see the questions in my above review.

---

> ### Author Response · Authors · 2025-11-21
>
> We thank Reviewer zYv3 for their review, and recognizing strengths in both the clarity of ideas and empirical performance. We include initial responses to the review comments below. We are also compiling additional results to further address concerns, we will follow up shortly with these. Please let us know if you have any questions or concerns in the meantime.
>
> > "Analysis that says when this bias is problematic"
>
> Thank you for this note; we will update our paper with more discussion on the bias introduced by the new ratio approximation. Mathematically, we note that this bias only appears for off-policy samples: Equations (8) and (9) are equal when the ratio is 1 or close to 1, which is enforced by the trust region constraint. Empirically from Figure 3, we observe that the bias introduced by computing the ratio on a per-sample basis primarily impacts the early stages of training. As shown in Figure 3, FPO++ with per-action ratio (the lower-bias formulation) initially trains faster, but the approach soon destabilizes and collapses across most tasks.
>
> On the concern that "if the samples are themselves high variance and have high dynamic range, then the gradient may actually vanish": while we haven't observed this problem in practice, we agree that this is conceptually possible and may be interesting to study in future work.
>
> > '[SPO] "also reduces gradient variance" and improves stability but again -- when is this problematic? Could the gradient vanish in this case too?'
>
> Vanishing gradients are directly mitigated by the SPO design. Unlike standard PPO clipping, which zeros out gradients outside the clip region (leading to gradient sparsity), the SPO objective retains a gradient signal for every sample. This reduces the overall variance of gradients. A more detailed analysis of SPO’s gradient behavior is provided in the Simple Policy Optimization (Xie et al.) paper.
>
> In FPO++, our main use of SPO is to preserve gradient signals for negative-advantage samples. Preserving gradient signals for negative advantages (ASPO) discourages over-aggressive likelihood decreases, which we find results in more stable policy learning. The improved entropy preservation is visualized in Figure 5; improvements in final training rewards are shown in Figure 4. To strengthen this claim further, we are also preparing and will follow up with quantitative results that show this improvement holds across more hyperparameter choices.
>
> > "How did the Huber kernel get chosen?"
>
> Thank you for asking this question; it was also raised by Reviewer YchB01. We used the standard Huber kernel value of 2 across environments: as stated on L178-179, this is primarily to avoid overflow from exponentiating squared values. The Huber loss only impacts outlier samples that are rare by construction, so the effect of the kernel choice ends up being very small. We will follow up with more quantitative results that sweep both the Huber loss and clamping parameters described at the end of Section 2.3.
>
> > "insight as to whether these changes in general are appropriate for most applications"
>
> Thank you for this suggestion. Our experience is that these contributions are general across tasks: to provide additional evidence, we have conducted new experiments applying FPO++ to a manipulation task that uses action chunking, where prior works (Diffusion Policy, Pi0) have validated the benefits of flow models. FPO++ shows favorable and stable performance in this new domain, demonstrating its ability to robustly handle policy tasks that rely on greater expressive capacity. We will follow up with more details on these results soon.
>
> > "the paper is not precise about what the benchmarks are -- the cited paper is for Orbit which is a simulation environment paper, not a benchmark paper."
>
> We believe the paper is already clear on this point: following Line 267 of the paper, we use the standard IsaacLab Go2, Spot, H1, and G1 velocity-conditioned locomotion tasks. Orbit, the standard citation for IsaacLab, is a simulation framework that proposes a set of benchmarks. From the Orbit paper abstract, Orbit "[provides] a suite of benchmark tasks of varying difficulty".
>
> > "Presumably the episode return in figures 2-6 are forward progress in some manner -- this is not stated."
>
> The goal of the episode returns in these figures is to highlight the relative impact of different implementation choices. In Figure 2, the episode return for FPO collapses, while FPO++ trains stably. Figure 3, 4, and 5 show the impact of individual algorithmic choices in FPO++: stability improvements from the updated ratio approximation, improved rewards from converged policies from the trust region method, and improved entropy preservation respectively. The goal of Figure 6 is to show that FPO++ can converge to higher training returns than a Gaussian PPO policy.

---

### Official Review · Reviewer_AXpp · 2025-10-30

**Soundness:** 2
**Presentation:** 3
**Contribution:** 3
**Rating:** 6
**Confidence:** 4

**Summary:**

This paper addresses challenges with the existing Flow Policy Optimization (FPO) and proposes improvements for successful deployment in real-world robotic experiments. The authors also include strong supplementary material demonstrating the policy’s performance on physical robots, a commendable and non-trivial effort.

They employ a "per-ratio clipping" mechanism and replace standard PPO with an entropy-preserving trust region (ASPO), claiming that these design choices are crucial for stable performance.

While the authors note that outperforming Gaussian PPO is not the primary goal, it is nonetheless used as the main baseline. This choice is somewhat questionable. Why not include more relevant baselines such as DIME [1] or DDPO [2]?

[1] DIME: Diffusion-Based Maximum Entropy Reinforcement Learning (Celik et al.)

[2] Training Diffusion Models with Reinforcement Learning (Black et al.)

**Strengths:**

- Impressive results on real robot experiments – a significant and practical achievement.
- The topic is relatively new; related work is mostly well-covered (though missing DIME).
- The paper is clearly written and easy to follow.

**Weaknesses:**

- The contribution is largely a combination of engineering and design choices rather than a fundamentally new method.
- The experimental analysis is underdeveloped. For example, Figure 3 shows that the per-sample ratio has a large effect, but the paper lacks further investigation into vanilla FPO (e.g., varying batch sizes, the number of monte carlo samples, gradient clipping, etc.) aside from a brief description in section 2.2.
- The claim of reduced gradient variance is made without theoretical or empirical evidence. The argument using Jensen’s inequality on averaged importance weights is questionable: maximizing an upper bound (on the weighted advantage) does not seem well-justified and may even be problematic from a theoretical standpoint. Since this is listed as a key contribution, it should be justified more rigorously.
- The paper seems to straddle two goals: (1) investigating methods for improving FPO, and (2) showcasing real-world robotics results. It would be stronger if it leaned more clearly toward one of these directions or made more comprehensive use of the appendix to address weaknesses.
- The relative importance of the three main components in Section 2.4 (“Final FPO++ Objective”) is not adequately analyzed in the results. As a smaller note, item 4 (“epsilon starting at zero”) does not seem distinctive enough to be listed as a key contribution.

**Questions:**

- Figure 5 (single action dimension) is not very informative, and the specific action dimension is not discussed. Showing the full action space would be more meaningful (e.g., including additional DoFs in the appendix, or projecting all dimensions to a 2D space somehow). The resulting distribution is unimodal, which weakens the motivation for using a more expressive flow-based policy over a Gaussian one. Do the authors observe any cases where a more complex action distribution emerges?
- Did the authors attempt adaptive learning rates (e.g., based on the trust region) or smaller Monte Carlo sample sizes (smaller N) to address FPO instability?

## Minor Comments
- Unclear notation in the legends of Figures 3 and 4. I assume “w/” is meant to indicate “without” (“w/out” would be clearer).
- The action correlation visualization is an interesting idea but lacks clear practical value. Similar correlations could likely appear with Gaussian PPO, so this figure might be better suited for the appendix.
- Reported return and episode length are difficult to interpret without context on the environment horizon or reward scale. Providing this context would improve clarity.


## Overall Assessment
The paper is valuable for its applicability and engineering contributions to real-world robot learning. However, its scientific claims are weakly supported, particularly regarding gradient variance reduction and the claimed expressivity benefits of flow-based policies. As a result, the theoretical and empirical analyses require further development. However, I commend the authors’ effort in real-world validation and supplementary videos, which are the clear strengths of the paper.

---

> ### Author Response · Authors · 2025-11-21
>
> We thank Reviewer AXpp30 for the constructive feedback and recognizing the significance of our practical results, as well as the clarity and presentation of the paper. We include initial responses below; we will also follow up with additional results.
>
> > Why not include more relevant baselines such as DIME or DDPO?
>
> We will update our related work section to discuss DIME (Celik et al) and DDPO (Black et al).
>
> We don't compare against these methods because they fall under different algorithm classes: FPO++ is an on-policy RL algorithm that we use to train policies from scratch, DIME is an off-policy RL algorithm, and DDPO is on-policy but used for fine-tuning existing image diffusion. Off-policy RL algorithms have seen limited success so far for real-world robot control; successfully adapting either DIME (off-policy RL) or DDPO (image diffusion fine-tuning) to our experiments would be a large endeavor in and of itself.
>
> Other baselines: the original FPO does compare against DPPO (Ren et al), but DPPO is also designed for fine-tuning. To our knowledge, no other methods have been shown to successfully sim-to-real flow-based control policies from scratch.
>
> >  "Further investigation into vanilla FPO", "adaptive learning rates", "Monte Carlo sample sizes"
>
> We will run hyperparameter sweeps to characterize the performance of vanilla FPO more concretely, and to provide detailed evidence beyond the observations presented in Section 2.2. We will follow up as soon as these results are ready.
>
> > "Reduced gradient variance [claimed] without theoretical or empirical evidence."
>
> Thank you for raising this point; it was also pointed out by Reviewer U4wE. To clarify, our discussion on gradient variance reduction aims to be an intuitive explanation for (i) Equation 9, which provides a finer-grained trust region, and (ii) the use of an SPO trust region. A finer-grained trust region makes it easier to clip outliers from Monte Carlo sampling, while the SPO objective increases the number of terms that are averaged over and contribute to the final gradient. These algorithmic choices are directly connected to gradient variance by construction: we will revisit the phrasing to make sure this clearer. We will also explore direct measurements for quantifying the variance reduction.
>
> > "Argument using Jensen’s inequality on averaged importance weights."
>
> We agree the inequality does not justify the correctness of the new ratio approximation. To clarify, the intent of the inequality (Equation 10) was not as justification for the ratio or evidence of reduced variance. Instead, it was to acknowledge a bias/tradeoff. The per-sample formulation introduces a positive bias for off-policy samples (for on-policy samples, Equation 8 and 9 have equal gradients); despite the bias, it improves training stability in Figure 2. We will revisit the writing to make sure these points are clearer.
>
> > "Paper seems to straddle two goals"
>
> We appreciate the reviewer's observation regarding the dual focus of our paper (algorithmic improvements and real-world results). To us, however, this approach is intentional and necessary. One takeaway of our experiments was that FPO's instabilities (Figure 2, Section 2.2) were not visible in simpler benchmarks but became pronounced when applied to higher-dimensional and more difficult robotics tasks. Improving FPO is what enables real-world robotics results, while the real-world robotics results provide evidence that the improvements are successful and meaningful.
>
> > "Relative importance of the three main components [...] not adequately analyzed"
>
> We will update our results section with more discussion about Figures 3, 4, and 5!
>
> > "Full action space would be more meaningful", "resulting distribution is unimodal", "cases where a more complex action distribution emerges?"
>
> We agree that the Figure 5 visualization can be improved; we will update the paper with more action dimension visualizations.
> Unimodality: the original FPO paper includes visualizations that demonstrate that more multimodal action distributions can emerge when flow policies are trained for tasks that benefit from them. For both practical and scientific reasons, we chose existing robot reinforcement learning tasks to evaluate FPO++. Unimodality is expected for these tasks because they were developed in frameworks that rely heavily on Gaussian PPO.
>
> To provide evidence that our conclusions hold for more complex action distributions, we have conducted new experiments applying FPO++ to a manipulation task that uses action chunking, where prior works (Diffusion Policy, Pi0) have validated the necessity of more expressive action distributions. FPO++ shows favorable performance compared to baselines in this new domain, demonstrating its ability to robustly handle policy tasks that rely on greater expressive capacity. We will follow up with more details on these results.

---

> ### Author Response · Authors · 2025-11-22
>
> To follow up on the minor comments:
>
> > 'I assume “w/” is meant to indicate “without” (“w/out” would be clearer).'
>
> "w/" is an abbreviation for "with", not without. Figure 4 aims to show that FPO++ (which uses a per-sample ratio, Equation 9) is significantly more stable than an alternate implementation that uses a per-action ratio ("w/ per-action", Equation 8).  Figure 5 aims to show that ASPO is important, and results in higher final training returns when compared to PPO and SPO.
>
> We will update the labels to make this clearer.
>
> > Similar correlations could likely appear with Gaussian PPO.
>
> We agree that these correlations are possible with full covariance Gaussian PPO, despite not being possible with the diagonal Gaussian PPO policies that are the standard practice in the field. We will update the caption with this context. We still believe that this visualization is useful: it shows that FPO++ is able to leverage the expressivity of the flow representation in meaningful ways.
>
>  > Reported return and episode length are difficult to interpret.
>
> We will add more details about the reward formulation and implementation details to the appendix. This will make these numbers easier to interpret.

---

> > ### Comment · Reviewer_AXpp · 2025-11-26
> >
> > Thank you for the clarifications. I will await the new changes to provide further feedback/questions. In the meantime, I have a couple of follow-up questions...
> >
> > 1) Could you please label the x-axis on the return plots in your paper? Furthermore, are these figures generated using the full Beyond Mimic dataset with the same domain randomization used to create the videos? If not, this should be clearly stated, and the training curves used for the real-world deployment should be presented.
> >
> > 2) Although the new manipulation experiments can help show multimodality, this feels like a large addition that is outside the scope of the original paper (flow policies for locomotion).

---

> > > ### Author Response · Authors · 2025-11-26
> > >
> > > Thank you for the reply and additional feedback. We will update you as soon as the additional results and changes are ready.
> > >
> > > Responses to the new questions:
> > > 1. *Return plots.* We will update all plots with x-axis labels (training step). Figures 2, 3, and 4 are generated using the IsaacLab locomotion environments discussed in Section 3.1 ("Locomotion Benchmarking"), not the BeyondMimic motion tracking task. We will add the BeyondMimic training curves with full domain randomization to the revision.
> > > 2. *Extra manipulation experiments.* Thanks for this feedback, we understand the concern about scope. To confirm, we don't plan to change the focus of the paper. Our investigation and real-world robot results will remain centered on legged robots. The goal of this supplementary addition is to highlight that the resulting algorithmic contributions have more general potential. Our hope is that this will be helpful for assessing the potential impact of our work.

---

### Official Review · Reviewer_U4wE · 2025-11-02

**Soundness:** 3
**Presentation:** 3
**Contribution:** 3
**Rating:** 4
**Confidence:** 3

**Summary:**

The paper proposes modifications to Flow Policy Optimization (FPO) to prevent instability and entropy collapse in robot control, and shows that the resulting FPO++ learns legged locomotion and motion tracking in simulation and transfers zero‑shot to two humanoid robots, albeit with limited baselines and generalisation.

**Strengths:**

- **Extensive Benchmarks:** The paper provides thorough experimental validation across four challenging locomotion tasks (Go2, Spot, H1, G1) in IsaacLab. Also, includes detailed ablation studies isolating each proposed component and direct comparisons against Gaussian PPO.

- **Empirical Results:** FPO++ demonstrates stable training without entropy collapse and consistently achieves higher or comparable returns relative to Gaussian PPO,.

- **Sim-to-Real Transfer:** Policies trained entirely in simulation transfer successfully to real humanoid robots, exhibiting robust walking and motion-tracking behaviors without additional fine-tuning.

**Weaknesses:**

- **Limited Baselines:** The paper mostly compares FPO++ with Gaussian PPO. It does not include other baselines like FPO or its variants. This makes it hard to know if the improvements come from the modifications to FPO.

- **Limited Evidence for Claimed Mechanisms:** The paper suggests FPO++ reduces variance and preserves entropy but does not provide direct measurements (e.g., gradient variance, policy entropy curves). The evidence is indirect, so the explanation is not fully convincing.

**Questions:**

Please see the weakness.

---

> ### Author Response · Authors · 2025-11-21
>
> We thank Reviewer U4wE for the positive assessment of our benchmarks, ablations, empirical results, and the significance of the sim-to-real transfer results. We also appreciate the feedback on (i) baselines and (ii) evidence for the claimed mechanisms, which we respond to below.
>
> > Limited baselines: “does not include other baselines like FPO”, "hard to know if the improvements come from the modifications to FPO"
>
> We would like to clarify that we our submission does directly compare FPO and FPO++: Figure 2 is exactly this comparison (FPO vs FPO++), while Figures 3, 4, and 5 isolate the effect of individual algorithmic differences between FPO and FPO++.
> We acknowledge, however, that we did not include quantitative vanilla FPO results for the sim-to-real tasks because it fails catastrophically. We can add these failure cases to our paper to make this clearer and reinforce the motivation of our work.
>
> To further strengthen this comparison and respond to concerns from Reviewer 1 (YchB), we also plan to add standard FPO results for a broader set of hyperparameters. We will follow up when these results are ready.
>
> > "Direct measurements (e.g., gradient variance, policy entropy curves)"
>
> We thank the reviewer for these suggestions. We would like to clarify these two points:
>
> - Our discussion on gradient variance reduction aims to be an intuitive explanation for (i) Equation 9, which provides a finer-grained trust region, and (ii) the use of an SPO trust region. A finer-grained trust region makes it easier to clip outliers from Monte Carlo sampling, while the SPO objective increases the number of terms that are averaged over and contribute to the final gradient. These algorithmic choices are directly connected to gradient variance by construction: we will revisit the phrasing to make sure this is not ambiguous.
> - We disagree with the characterization that the entropy preservation evidence is indirect: we include flow field visualizations in Figure 5, where higher-entropy action distributions from ASPO-trained policies are directly visible.
>
> Overall, however, we agree with the reviewer that more measurements would strengthen our paper. We are working on these measurements and will follow up when they are ready!

---

### Official Review · Reviewer_YchB · 2025-11-02

**Soundness:** 3
**Presentation:** 2
**Contribution:** 2
**Rating:** 2
**Confidence:** 4

**Summary:**

This paper extends flow policy optimization (FPO) to complex continuous control tasks like legged locomotion. Specifically, the authors empirically find 2 challenges in the existing FPO method. They observe that as the training progresses, the policy either collapses during training or there's a gradual decay once it reaches the peak. To address this, the authors propose to reduce the variance of gradients during training by computing the surrogate ratio per-sample instead of per-action, regularizing the entropy of actions with Asymmetric SPO. They show strong sim2real experiments on two humanoid robots for locomotion and whole body motion tracking.

**Strengths:**

1. The proposed method shows strong Sim2Real results (on multiple embodiments - Booster and Unitree G1) when compared to the FPO algorithm, which cannot be transferred to the real world for legged robots.

**Weaknesses:**

1. The motivation of "Policy Collapse during training" is lacking concrete experimentation. The authors on L124 say "even after tuning hyperparameters like learning rate, clip threshold, ..." but provide no plots backing up their claim in the main paper / in the appendix.

	- What normalization strategies have the authors evaluated to back up this claim?


2. I would like to see a comparison with state-of-the-art practices for Deep RL. Specifically:
	- Instead of an MLP, use a more BroNet/SimBa(v1/v2)-like architecture, which has skip connections + normalizations. There is strong evidence to suggest that adopting such architecture leads to much stable training.
	- Adopt RSNorm for the observation to be able to deal with non-stationarity in Deep RL appropriately.

3. [Unverified Huber loss experiments]: On L189, the authors mention that "We empirically verify the importance of both the Huber kernel and clamping in our experiments." However, there are no experiments showing this.
	- Does the same value of $\delta$ work across different environments?
	- How sensitive is the method ot $\delta$?

4. Figure 4 shows that FPO++ without the ASPO objective -- either with PPO or SPO-- is as performant as the proposed FPO++. It is unclear, then, as to what ASPO's contribution really is. Can the authors elaborate on this?


5. The hyperparameters section in the Appendix is quite small and makes me question the reproducibility of this paper. No mention of the huber and clamp parameters (unless they are the same as the clip parameters -- but then they also need to be validated with a hyperparameter sweep), number of environments used, mini-batch size during the training, etc.

**Questions:**

6. How significant is the difference between FPO++ and PPO when it comes to Table 1 (Motion tacking)?

---

> ### Author Response · Authors · 2025-11-21
>
> We thank Reviewer YchB for the detailed review and for recognizing the strong sim-to-real results on multiple embodiments. We appreciate your constructive feedback and are including initial responses to your concerns below. We are also compiling additional results in response to your feedback and will follow up as soon as these are ready. Please let us know if you have any further questions in the meantime.
>
> > "The motivation of "Policy Collapse during training" is lacking concrete experimentation." Does FPO really fail?
>
> Thank you for raising this. We have observed that standard FPO policies collapse consistently for the tasks in our paper, and we will run more sweeps to characterize this. This has also been observed in external concurrent work that re-implements FPO: [link](https://github.com/vruga/lerobot-sim2real/blob/e222b53699bb325d160c471b9ae3497a1e048ae5/docs/evaluations.md), where we observe similar collapsing patterns; we have also confirmed implementation details of our method with the original authors of FPO. We will follow up with results from our hyperparameter sweep and make this motivation more clear in the paper.
>
> > "The hyperparameters section in the Appendix is quite small"
>
> We agree that hyperparameters are important, particularly for reinforcement learning. We will update the Appendix to include all hyperparameters and implementation details needed for reproducing the results of our paper; for reproducibility, we will also release code upon publication. In addition to improving reproducibility, we believe that more detailed implementation and hyperparameter information would have clarified other concerns raised by the reviewer, notably:
> - Normalization: following the original FPO implementation, we already apply the standard training techniques used in online RL for robot control. This includes running normalization for observations (similar to RSNorm), advantage normalization, and other stability tricks like gradient clipping and weight decay. We apply these techniques to all experiments in our paper: FPO++, FPO, and Gaussian PPO.
> - Architectural comparisons (BroNet/SimBa): we will add discussion to our paper on BroNet and SimBA, which we believe are highly relevant but not fully applicable to our setting. The focus of our work is on-policy learning for real-time continuous control, where networks are typically small and shallow. FPO++, FPO, and Gaussian PPO experiments in our paper use 256 unit, 3-layer policy MLPs for locomotion tasks and (1024, 512, 256) unit, 3-layer MLPs for motion tracking task. While BroNet and SimBA are relevant because they offer evidence that techniques like observation normalization are critical, they are primarily focused on training stability for much deeper architectures.
>
> > "Unverified Huber loss", Does the same value of $\delta$ work across different environments? How sensitive is the method to $\delta$?
>
> We used the same $\delta$ value across environments: the Huber loss only impacts outlier samples that are rare by construction, so the effect of the specific choice in $\delta$ ends up being very small. We will follow up with more quantitative results that sweep both the Huber loss and clamping parameters described at the end of Section 2.3.
>
> > "FPO++ without the ASPO objective -- either with PPO or SPO -- is as performant as the proposed FPO++."
>
> We observed significant and consistent improvements in converged policies from ASPO. While this can be observed on the right side of Figure 4, where FPO++ has the highest final reward in all tasks, we recognize the visual effect may be subtle. We are formatting and will follow up with quantitative results that show this improvement more definitively.
>
> > How significant is the difference between FPO++ and PPO when it comes to Table 1 (Motion tracking)?
>
> Qualitatively, we found that flow policies trained using FPO++ are able to achieve effectively the same performance as a policy trained using a state-of-the-art PPO implementation. Both methods demonstrated strong sim-to-real transfer for the dynamic motion sequence used in the real-world result shown in our anonymous website, maintaining the motion and rejecting perturbations for the full sequence duration of 2 minutes and 30 seconds.

---

### Author Response · Authors · 2025-11-30

We thank the reviewers for their constructive feedback. We are encouraged by recognition of contributions from our work: strengths listed by reviewers include "strong sim2real results" (R1), "extensive benchmarks" and "detailed ablation studies" (R2), that the paper "empirically delivers on the promised performance" (R4), and the "impressive results on real robot experiments", described as "a significant and practical achievement" (R3). Reviewers also highlighted clarity, for example, that the paper is "clearly written and easy to follow" (R3).

**Comments from reviewers align with what we see as the core contributions of our work**: (i) showing how flow policy gradients can be made effective for challenging control tasks where the existing FPO algorithm fails, and (ii) demonstrating for the first time that flow RL can be practical for dynamic control problems on real robots. We believe that these are important results for the robot learning community, particularly given the current interest in flow and diffusion models for humanoid control and manipulation.

We are also grateful for questions raised by reviewers. We have made clarifications or updates in response to key concerns from every review. To make this easier to verify, we consolidated concerns raised by each reviewer and the specific clarifications and changes we have made in response. New experiments and figures are added to the Appendix of our manuscript to keep figure numbering consistent with the submission, but we will incorporate results into the main paper body for the final draft.

We organize responses by reviewer below.

---

## R1:

1. *Submission claims that FPO fails "even after tuning hyperparameters", "but [authors] provide no plots backing up their claim".*
   - **New result.** We have run a comprehensive hyperparameter sweep (50+ experiments per environment) over FPO hyperparameters for all locomotion environments, and updated Figure 2 appropriately. The updated result shows that vanilla FPO's worse performance is more fundamental than poor hyperparameter tuning.

2. *Comparisons with "BroNet/SimBa(v1/v2)", and "adopt RSNorm for the observation".*
   - **Clarification.** We added discussion on BroNet and SimBa on L243-252. We also added more implementation details in Appendix C, which clarify that we are already using the standard techniques for observation normalization.

3. *"Unverified Huber loss"*
   - **New result.** We have added Figure A.2, which verifies the importance of the Huber loss and gradient-preserving clamping applied by FPO++ across all 4 robot locomotion environments. We observe that similar delta values succeed in all environments.

4. *"Unclear [...] what ASPO's contribution really is."*
   - **Clarification.** Figure 4 in the original submission shows that ASPO trains more stably than PPO, where performance decays after peaking for the H1 and G1 robots, while also training much more quickly than SPO. For the H1 robot, the final training returns are >10% higher for ASPO than PPO and SPO.
   - **New result.** We added Figure A.4, which provides further evidence for the contribution of the ASPO trust region. This figure shows that ASPO produces higher training and evaluation returns than PPO and SPO for many different clipping parameter and seed choices.
   - **New result.** We added Figure A.5, which verifies that the ASPO trust region reduces empirical gradient variance.

5. *"Hyperparameters section in the Appendix is quite small."*
   - **New result.** We have updated Appendix C of the manuscript with extensive implementation and hyperparameter details.

*(continued)*

---

> ### Author Response · Authors · 2025-11-30
>
> ## R2:
>
> 1. *Experiments do not "include other baselines like FPO or its variants."*
>    - **Clarification.** This is a misunderstanding of the submission's experiments. Figure 2 directly compares FPO++ against FPO, while Figures 3, 4, and 5 isolate the effect of algorithmic differences between FPO and FPO++.
>
> 2. *"Limited evidence for claimed mechanisms" (entropy preservation, reduced gradient variance).*
>    - **Clarification.** Figure 5 of the submission shows flow field visualizations that verify improved entropy preservation from the ASPO trust region.
>    - **New result.** We have added visualizations for verifying entropy characteristics for additional action dimensions in Figure A.7, A.8, A.9. These visuals show low entropy during policy collapse with a PPO trust region, and entropy preserved using the ASPO trust region.
>    - **New result.** We have added gradient variance measurements in Figure A.5, which adapts the technique from "A Closer Look at Deep Policy Gradients" (Ilyas et al, ICLR 2020) of using cosine similarity to measure gradient variance. We observe that the ASPO trust region and per-sample ratio estimate increase cosine similarity between individual and averaged gradients; this provides evidence for lower gradient variance.
>
> ## R3:
>
> 1. *"Paper lacks further investigation into vanilla FPO." "Did the authors attempt adaptive learning rates", "smaller Monte Carlo sample sizes"?*
>    - **New result.** We have updated Figure 2 of the paper with training curves from an extensive FPO hyperparameter sweep. See response to concern 1 from R1.
>
> 2. *"Claim of reduced gradient variance is made without theoretical or empirical evidence."*
>    - **New result.** We have added empirical evidence of reduced gradient variance. See response to concern 2 from R2.
>
> 3. *"Argument using Jensen's inequality as justification is questionable."*
>    - **Clarification.** This inequality was written to acknowledge a tradeoff, not to justify the method. We have updated this section of the paper.
>
> 4. *"Item 4 ('epsilon starting at zero') does not seem distinctive enough."*
>    - **New result.** We have removed this item from the referenced list. We still believe, however, that this detail is important and are still discussing it on Lines 445-450. We have also added empirical evidence for the value of zero initialization in Table A.3 and A.4 in Appendix F, where we find it produces significant increases to evaluation returns (>20% for motion tracking).
>
> 5. *On flow field visualizations, "showing the full action space would be more meaningful".*
>    - **New result.** We have added visualizations for every action dimension in Figures A.7, A.8, and A.9. These visualizations are consistent with the single-dimension visualization in the original submission.
>
> ## R4:
>
> 1. *"How did the Huber kernel get chosen?"*
>    - **New result.** We have added a figure with our sweep over the Huber kernel parameter, which generalizes across tasks. See response to concern 3 from R1.
>
> 2. *[Initializing epsilon at 0] "seems to be a fairly minor step."*
>    - **New result.** We have added new results. See response to concern 4 from R3.
>
> 3. *"Whether these changes in general are appropriate for most applications [...] or if they really are specific to legged locomotion"*
>    - **New result.** We have added results on manipulation tasks in Appendix E. FPO++'s advantages also apply here, which demonstrates generality.
>    - **Context.** As the reviewer acknowledges, our submission already validated that FPO++ generalizes across 5 different legged robot embodiments.
>
> 4. *"Paper is not precise about what the benchmarks are -- the cited paper is for Orbit which is a simulation environment paper, not a benchmark paper"*
>    - **Clarification.** We believe the paper is already clear: we use the standard IsaacLab locomotion tasks. Orbit, the standard citation for IsaacLab, is a simulation framework that proposes a set of benchmarks. From the Orbit paper abstract, Orbit "[provides] a suite of benchmark tasks of varying difficulty".
>
> 5. *"The results in figures 7-9 are impressive… do not necessarily [convince] us that the authors have overcome some issues with numerical conditioning."*
>    - **Clarification.** The goal of Figure 7 is indeed not to provide evidence of numerical conditioning fixes. Instead, it is simply to highlight that FPO++ training successfully leverages the expressivity of the flow representation, in ways that are not possible using standard diagonal Gaussian policies.
>    - **Clarification.** We respectfully disagree with this characterization for Figures 8 and 9. Together with Figure A.6, these results show that FPO++ enables successful flow policies on challenging, domain-randomized real-world robot tasks. We view this as a significant result and evidence that FPO++ meaningfully resolves the failure modes of vanilla FPO.
>
> ---
>
> We hope this summary clarifies our contributions, and we thank the reviewers and AC for their time.

---

### Meta-Review · Area_Chair_z3wE · 2026-01-01

**Summary:**

While the paper demonstrates strong real-robot results, the core contribution is largely a collection of engineering choices with limited theoretical justification. Despite additional experiments in the rebuttal, key concerns remain regarding insufficient analysis of trade-offs, unclear generality beyond the evaluated settings, and weak support for variance-reduction and expressivity claims. As a result, the work does not yet meet the expected level of scientific rigour for acceptance.

**Reviewer Concerns:**

The rebuttal added clarifications and new experiments on hyperparameter sensitivity, vanilla FPO behaviour, entropy preservation, and gradient variance, and improved implementation and reproducibility details. However, the key concerns remain about limited theoretical justification, insufficient analysis of trade-offs and failure modes of the proposed components, unclear generality beyond legged locomotion, and the largely engineering-driven nature of the contribution.

**Reviewer Scores:**

Overall, the reviewers’ concerns about the weaknesses are valid, and the authors’ rebuttal does not adequately address these issues.

---

### Decision · Program_Chairs · 2026-01-26

Reject